# METTL3-dependent m$^6$A modification programs T follicular helper cell differentiation

Yingpeng Yao [1,5], Ying Yang[2,3,5], Wenhui Guo[1,5], Lifan Xu[4,5], Menghao You[1], Yi-Chang Zhang[2,3], Zhen Sun[1], Xiao Cui[1], Guotao Yu[1], Zhihong Qi[1], Jingjing Liu [1], Fang Wang[1], Juanjuan Liu[1], Tianyan Zhao[1], Lilin Ye[4✉], Yun-Gui Yang [2,3✉] & Shuyang Yu [1✉]

T follicular helper (T$_{FH}$) cells are specialized effector CD4$^+$ T cells critical to humoral immunity. Whether post-transcriptional regulation has a function in T$_{FH}$ cells is unknown. Here, we show conditional deletion of METTL3 (a methyltransferase catalyzing mRNA $N^6$-methyladenosine (m$^6$A) modification) in CD4$^+$ T cells impairs T$_{FH}$ differentiation and germinal center responses in a cell-intrinsic manner in mice. METTL3 is necessary for expression of important T$_{FH}$ signature genes, including *Tcf7*, *Bcl6*, *Icos* and *Cxcr5* and these effects depend on intact methyltransferase activity. m$^6$A-miCLIP-seq shows the 3′ UTR of *Tcf7* mRNA is subjected to METTL3-dependent m$^6$A modification. Loss of METTL3 or mutation of the *Tcf7* 3′ UTR m$^6$A site results in accelerated decay of *Tcf7* transcripts. Importantly, ectopic expression of TCF-1 (encoded by *Tcf7*) rectifies T$_{FH}$ defects owing to METTL3 deficiency. Our findings indicate that METTL3 stabilizes *Tcf7* transcripts via m$^6$A modification to ensure activation of a T$_{FH}$ transcriptional program, indicating a pivotal function of post-transcriptional regulation in promoting T$_{FH}$ cell differentiation.

[1] State Key Laboratory of Agrobiotechnology, College of Biological Sciences, China Agricultural University, Beijing, China. [2] CAS Key Laboratory of Genomic and Precision Medicine, Collaborative Innovation Center of Genetics and Development, CAS Center for Excellence in Molecular Cell Science, College of Future Technology, Beijing Institute of Genomics, Chinese Academy of Sciences, Beijing, China. [3] University of Chinese Academy of Sciences, Beijing, China. [4] Institute of Immunology, Third Military Medical University, Chongqing, China. [5] These authors contributed equally: Yingpeng Yao, Ying Yang, Wenhui Guo, Lifan Xu. ✉email: yelilinlcmv@tmmu.edu.cn; ygyang@big.ac.cn; ysy@cau.edu.cn

The production of high-affinity antibodies and generation of memory cells are critical for establishing protective immunity[1,2]. During acute viral infection, T follicular helper ($T_{FH}$) cells provide help to cognate antigen-presenting B cells to facilitate the formation of germinal centers (GCs) and the development of long-lived plasma cells and memory B cells[3]. After priming by dendritic cells (DCs) in the T-cell zone, $CD4^+$ T cells upregulate the expression of chemokine receptor CXCR5, the costimulatory receptor ICOS and the transcriptional repressor B-cell lymphoma 6 (Bcl-6), which together coordinate early fate commitment of $T_{FH}$ cells and their migration to the follicles. After interacting with cognate antigen-presenting B cells, these activated $CD4^+$ T cells mature into $T_{FH}$ cells and further differentiate into GC $T_{FH}$ cells, providing help to GC B cells for humoral immunity[4,5].

$T_{FH}$ differentiation is a complicated biological process[6], which is tightly controlled by multiple transcription factors. As a master regulator of $T_{FH}$ differentiation, Bcl-6 represses the $T_H1$, $T_H2$, and $T_H17$ lineage-specific transcription factors to promote the differentiation of activated $CD4^+$ T cells into $T_{FH}$ cells[7–9]. On the contrary, Blimp1 (encoded by $Prdm1$) negatively modulates Bcl-6 transcription by binding to the $Bcl6$ promoter and acting as a repressor, thus preventing $T_{FH}$ differentiation and promoting the formation of non-$T_{FH}$ effector helper T cells[9,10]. Accumulative studies have demonstrated that TCF-1 (encoded by $Tcf7$) plays key roles in $T_{FH}$ differentiation by directly acting upstream of the Bcl-6–Blimp1 axis[11–13]. Although the regulation of these transcription factors has been intensely investigated at the transcriptional level, it remains unknown if post transcriptional mechanisms are involved in the balanced expression of these key factors during $T_{FH}$ differentiation.

$N^6$-methyladenosine (m6A) is the most prevalent modification on eukaryote mRNAs, catalyzed by m6A methyltransferases[14]. Methyltransferase like 3 protein (METTL3, encoded by $Mettl3$) is the core catalytic subunit of m6A methyltransferases, and METTL14 is an allosteric activator of METTL3[15,16]. The recruitment of m6A-binding proteins is actively involved in almost every stage of mRNA metabolism, from processing in the nucleus to translation and decay in the cytoplasm, adding another layer of regulatory mechanisms in gene expression[17]. Early studies showed that m6A-tagged transcripts had shorter half-life[18], and binding of YTHDF2 to m6A promoted m6A-modified mRNA decay[18,19]. Recently, it was discovered that a group of novel m6A-binding proteins, IGF2BPs, stabilized methylated transcripts by guarding them against degradation[20,21]. The distinct roles of YTHDF2 and IGF2BPs suggest that the impact of m6A modification on mRNA stability is highly dependent on cell context.

Several studies have uncovered critical roles of m6A modification in immune regulation. It has been documented that m6A methylation is essential for normal hematopoiesis and leukemia development[22–24]. m6A methylation regulates T-cell homeostasis, as well as suppressive functions of Treg cells by targeting $Socs$ genes[25,26]. In the context of viral infection, m6A represses type I interferon production in an innate antiviral state[27,28]. Despite these profound effects of m6A modification on immunoregulation, its role in $T_{FH}$ differentiation has not been determined.

In this study, by conditional targeting the $Mettl3$ gene in T cells, we demonstrate that METTL3-mediated m6A modification is critical for $T_{FH}$ cell differentiation. Mechanistically, METTL3 deficiency impairs the stability of m6A-modified $Tcf7$ mRNA, resulting in compromised activation of $T_{FH}$ transcriptional program.

## Results

### METTL3 controls $T_{FH}$ differentiation and GC reactions.
To investigate the role of METTL3 in $T_{FH}$ differentiation, $Mettl3$-floxed mice were crossed with $Cd4$-Cre mice to generate conditional deletion of METTL3 in T cells ($Mettl3^{fl/fl}Cd4$-Cre mice), and the deletion efficiency of METTL3 in splenic $CD4^+$ T cells was validated by quantitative RT-PCR (Supplementary Fig. 1a). We then infected $Mettl3^{fl/fl}Cd4$-Cre mice and their wild-type control littermates (Ctrl) with LCMV-Armstrong strain. The frequency and numbers of $CD44^+CXCR5^+$ $T_{FH}$ cells were significantly diminished in $Mettl3^{fl/fl}Cd4$-Cre mice compared with those in their control littermates on day 8 post viral infection (Fig. 1a, b). In contrast, METTL3-deficient $CD4^+$ T cells were dramatically skewed toward the $CD44^+CXCR5^-$ $T_H1$ proportion, albeit the numbers of $T_H1$ cells were also decreased with the ablation of METTL3 (Fig. 1a, b). It is worth mentioning that ablation of METTL3 resulted in more severe defects in $T_{FH}$ cells (21.4-fold change) than $T_H1$ cells (1.9-fold change) upon acute viral infection (Fig. 1b). Given T-bet is expressed at a relatively high level on $T_H1$ lineage and directs its commitment[29], we also analyzed T-bet expression on both $T_H1$ and $T_{FH}$ cells. We found $T_H1$ cells expressed a much higher level of T-bet than $T_{FH}$ cells, and both $T_{FH}$ and $T_H1$ cells downregulated T-bet expression in the absence of METTL3 (Supplementary Fig. 1b). Moreover, $Mettl3^{fl/fl}Cd4$-Cre mice exhibited remarkably lower percentages and absolute cell numbers of GC $T_{FH}$ cells (identified as PD-$1^{hi}CXCR5^+$, $ICOS^{hi}CXCR5^+$, or $Bcl$-$6^{hi}CXCR5^+$) than those of their control littermates (Fig. 1c, d). Accordingly, the expression levels of CXCR5, PD-1, ICOS, and Bcl-6 were dramatically lower on METTL3-deficient $T_{FH}$ cells than on wild-type cells (Supplementary Fig. 1c, d). These results indicated that METTL3 deficiency severely impairs $T_{FH}$ differentiation.

Given the main function of $T_{FH}$ cells is to provide cognate B-cell help, which is a fundamental aspect of humoral immunity and generation of immunological memory[1]. We next examined whether METTL3 deficiency in $CD4^+$ T cells affects GC formation during viral infection. A robust reduction in the proportions and numbers of $GL$-$7^+Fas^+$ GC B cells (Fig. 1e, f) and $PNA^+Fas^+$ GC B cells (Supplementary Fig. 1e, f) were observed in $Mettl3^{fl/fl}Cd4$-Cre mice, compared with those in their wild-type counterparts. Furthermore, the frequency and cell numbers of $IgD^{lo}CD138^+$ plasma cells were also much lower in $Mettl3^{fl/fl}Cd4$-Cre mice than those in wild-type mice (Fig. 1e, f). The immunohistochemical analysis further confirmed substantially reduced $PNA^+$ GCs within B-cell follicles in spleens from $Mettl3^{fl/fl}Cd4$-Cre mice (Fig. 1g). To assess the consequences of defective $T_{FH}$ and GC responses in $Mettl3^{fl/fl}Cd4$-Cre mice, we measured the LCMV-specific serum concentration of immunoglobulin G (IgG). The LCMV-specific IgG concentration was significantly lower in $Mettl3^{fl/fl}Cd4$-Cre mice than that in their wild-type control littermates on day 8 and day 56 post viral infection (Fig. 1h). Collectively, these results suggested that METTL3 is required for $T_{FH}$ differentiation and GC responses.

To further validate these findings, we immunized $Mettl3^{fl/fl}Cd4$-Cre mice and their wild-type control littermates by intraperitoneal administration of keyhole limpet hemocyanin (KLH) emulsified in Complete Freund's Adjuvant (CFA). On day 8 post immunization, $Mettl3^{fl/fl}Cd4$-Cre mice exhibited impaired development of $T_{FH}$ cells and GC $T_{FH}$ cells, but not $T_H1$ cells (Supplementary Fig. 2a, b). Meanwhile, the percentages and cell numbers of GC B cells and plasma cells (Supplementary Fig. 2c, d) were also impaired in $Mettl3^{fl/fl}Cd4$-Cre mice compared with their wild-type counterparts. These data jointly indicated METTL3 promotes $T_{FH}$ differentiation upon different antigen stimulation.

Meanwhile, the differentiation of other T helper lineages was also examined by using KLH immunization model. We found that both $GATA3^+$ and IL-4-producing cells, as well as $Foxp3^+$ cells were not altered in $Mettl3^{fl/fl}Cd4$-Cre mice (Supplementary

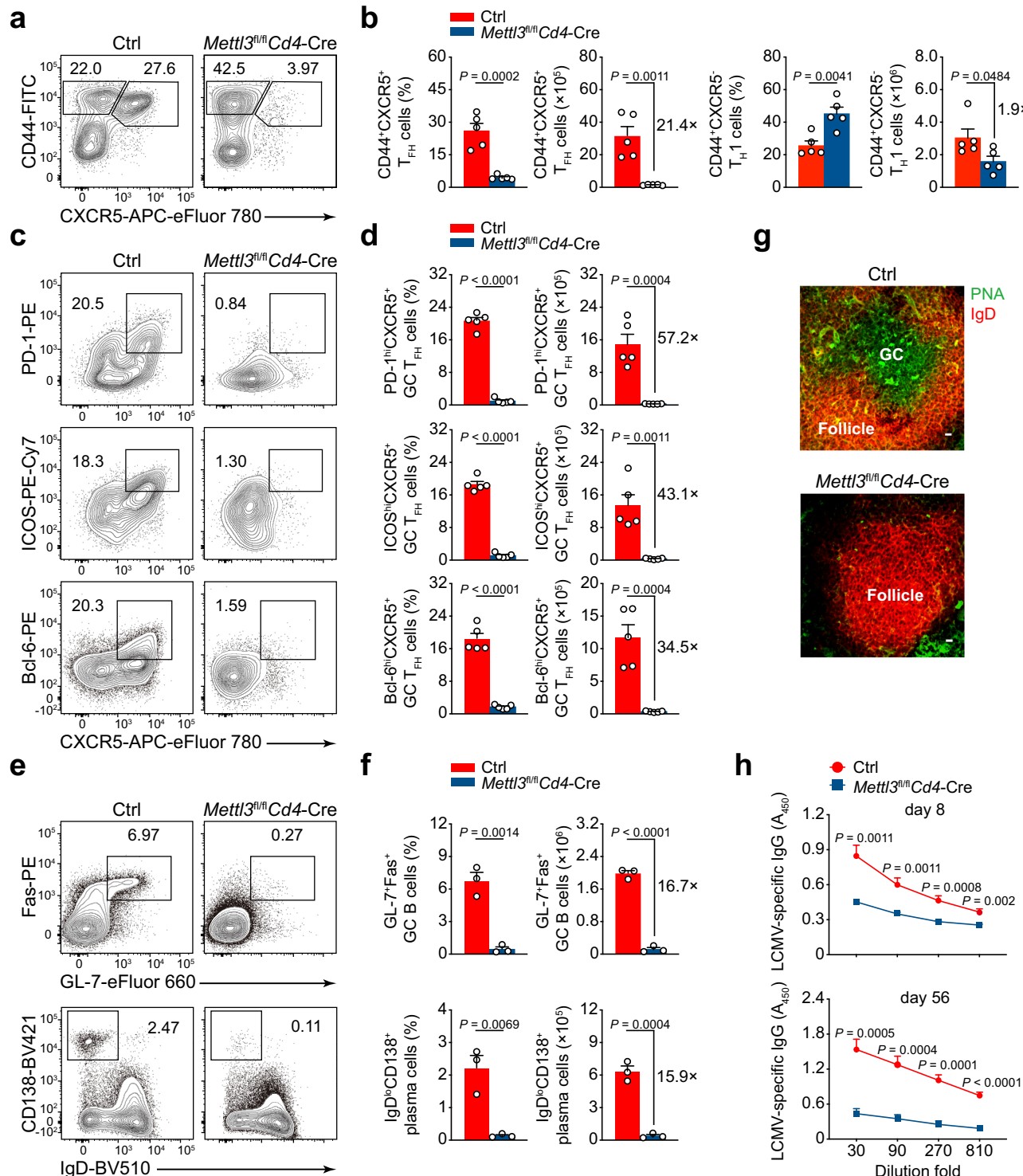

**Fig. 1 Conditional ablation of METTL3 impairs $T_{FH}$ differentiation and GC responses. a, b** Flow cytometry analysis of $CD44^+CXCR5^+$ $T_{FH}$ cells and $CD44^+CXCR5^-$ $T_H1$ cells, gated on splenic $CD4^+$ T cells from Ctrl and *Mettl3*^fl/fl^*Cd4*-Cre mice on day 8 post infection (8 *dpi*). Summary of the frequency and cell numbers of indicated cell subsets are shown in **b** (*n* = 5 per group). **c, d** Flow cytometry analysis of $PD-1^{hi}CXCR5^+$ GC $T_{FH}$ cells (top panel), $ICOS^{hi}CXCR5^+$ GC $T_{FH}$ cells (middle panel), and $Bcl-6^{hi}CXCR5^+$ GC $T_{FH}$ cells (bottom panel), gated on splenic $CD44^{hi}CD62L^{lo}CD4^+$ T cells from Ctrl and *Mettl3*^fl/fl^*Cd4*-Cre mice on 8 *dpi*. Summary of the frequency and cell numbers of indicated cell subsets are shown in **d** (*n* = 5 per group). **e, f** Flow cytometry analysis of splenic $GL-7^+Fas^+$ GC B cells (top panel) and $IgD^{lo}CD138^+$ plasma cells (bottom panel) on 8 *dpi*. Summary of the frequency and cell numbers of GC B cells and plasma cells are shown in **f** (*n* = 3 per group). **g** Immunofluorescent staining of spleens from Ctrl and *Mettl3*^fl/fl^*Cd4*-Cre on 8 *dpi*. Green: PNA; Red: IgD; scale bar: 10 μm. **h** Analysis of LCMV-specific IgG concentration in serum on 8 *dpi* (top) and on 56 *dpi* (bottom) by ELISA (day 8: *n* = 12 for Ctrl group, *n* = 10 for *Mettl3*^fl/fl^*Cd4*-Cre group; day 56: *n* = 5 per group). Data are representative of at least three independent experiments. Error bars indicate standard error of the mean. *P* value was calculated by unpaired two-tailed Student's *t* test.

Fig. 2e–h), indicating $T_H2$ and Treg cell differentiation are not affected in the absence of METTL3 upon KLH immunization. Interestingly, both RORγt[+] and IL-17a-producing cells were significantly decreased in METTL3-deficient mice (Supplementary Fig. 2e–h), revealing METTL3 is essential for $T_H17$ cell differentiation in vivo upon protein immunization.

**METTL3 intrinsically regulates $T_{FH}$ differentiation.** We next focused on whether METTL3 regulates $T_{FH}$ differentiation in a cell-intrinsic manner using both bone marrow chimeric and adoptive transfer mice models. We first generated bone marrow chimeric mice by reconstituting lethally irradiated wild-type recipient mice (CD45.1[+]) with a mixture donor of bone marrow cells from *Mettl3*[fl/fl]*Cd4*-Cre mice (CD45.2[+]) and wild-type (CD45.1[+]) competitor mice (Supplementary Fig. 3a). On day 8 post LCMV-Armstrong infection, *Mettl3*[fl/fl]*Cd4*-Cre mice-derived CD4[+] T cells had much lower cell numbers of CD44[+]CXCR5[+] $T_{FH}$ cells than those of competitor bone marrow-derived CD4[+] T cells (Supplementary Fig. 3b, c). Correspondingly, CD4[+] T cells originated in *Mettl3*[fl/fl]*Cd4*-Cre mice also exhibited a reduction of PD-1[hi]CXCR5[+] GC $T_{FH}$ cells (Supplementary Fig. 3b, c). The expression levels of CXCR5, PD-1, ICOS, and Bcl-6 were significantly lower on METTL3-deficient $T_{FH}$ cells than those on competitor cells (Supplementary Fig. 3d).

To further rule out the effect of external factors on $T_{FH}$ differentiation, we adoptively transferred CD45.1[+] SMARTA CD4[+] T cells into CD45.2[+] *Mettl3*[fl/fl]*Cd4*-Cre or their wild-type control littermates (Fig. 2a). On day 8 post viral infection, we observed the percentages and numbers of CD44[+]CXCR5[+] $T_{FH}$ cells derived from CD45.1[+] SMARTA CD4[+] T cells were similar in the *Mettl3*[fl/fl]*Cd4*-Cre mice and their wild-type counterparts (Fig. 2b, c), indicating the microenvironment in the *Mettl3*[fl/fl]*Cd4*-Cre mice did not impair $T_{FH}$ differentiation. In contrast, METTL3-deficient CD4[+] T cells displayed compromised $T_{FH}$ differentiation compared with that of congenic wild-type CD4[+] T cells (Fig. 2b, c). Correspondingly, METTL3-deficient CD4[+] T cells also exhibited defects in their ability to differentiate into CD44[+]CXCR5[−] $T_H1$ cells (Fig. 2b, c). Besides, transferred CD45.1[+] SMARTA cells also profoundly promoted the GC B and plasma cells differentiation (Fig. 2d, e), GC formation (Fig. 2f), and excessive production of LCMV-specific IgG antibody (Fig. 2g) in *Mettl3*[fl/fl]*Cd4*-Cre recipients. Together, our data demonstrated METTL3 is intrinsically required for $T_{FH}$ cell differentiation and functions as a key role in GCs formation.

**METTL3 is essential for the early initiation of $T_{FH}$ cells.** During acute viral infection, effector CD4[+] T cells' commitment to the $T_{FH}$ lineage or $T_H1$ lineage emerges before the initiation of GCs[12]. To investigate whether METTL3 is required for early $T_{FH}$ specification in vivo, naive Ctrl or *Mettl3*[fl/fl]*Cd4*-Cre SMARTA CD4[+] T cells were labeled with cell-trace violet (CTV) and adoptively transferred into congenic recipient mice, followed by LCMV-Armstrong infection (Supplementary Fig. 4a). On day 3 post viral infection, SMARTA CD4[+] T cells of both genotypes showed similar upregulation of T-cell activation markers CD69 and CD44, and downregulation of CD62L (Supplementary Fig. 4b, c). In vitro cell culture analysis also revealed that *Mettl3*[fl/fl]*Cd4*-Cre SMARTA CD4[+] T cells displayed no obvious defects in T-cell activation (Supplementary Fig. 4d). Ctrl SMARTA CD4[+] T cells exhibited vigorously proliferation post viral infection, whereas METTL3-deficient cells displayed a delayed proliferation (Fig. 3a). Lineage commitment to $T_{FH}$ lineage has occurred on day 3 post viral infection, as detected by Bcl-6[+]CXCR5[+] cells[30]. Consistently, wild-type SMARTA cells developed robust numbers of Bcl-6[+]CXCR5[+] $T_{FH}$ cells, whereas METTL3-deficient cells

exhibited much lower percentages and cell numbers of Bcl-6[+] CXCR5[+] $T_{FH}$ cells (Fig. 3b). Tracking the cell division showed *Mettl3*[fl/fl]*Cd4*-Cre SMARTA cells were predominantly in third and fourth divisions, while Ctrl SMARTA cells had advanced to fifth and sixth divisions (Fig. 3c, d), reflecting the impaired proliferation of activated CD4[+] T cells in the absence of METTL3. Consistently, *Mettl3*[fl/fl]*Cd4*-Cre SMARTA cells exhibited a reduction of proportions and cell numbers of Bcl-6[+]CXCR5[+] $T_{FH}$ cells in indicated cell divisions compared with that of Ctrl SMARTA cells (Fig. 3e, f). Interestingly, as a critical regulator for early $T_{FH}$ differentiation[31], TCF-1 expression in CXCR5[+] cells was also dramatically reduced due to ablation of METTL3 (Fig. 3g, h). Moreover, we also observed compromised CD25[−]CXCR5[+] $T_{FH}$ cells in *Mettl3*[fl/fl]*Cd4*-Cre SMARTA cells by using the combination of CXCR5 and CD25 to identify $T_{FH}$ cells in activated CD4[+] T cells (Supplementary Fig. 4e, f), and apoptosis of $T_{FH}$ cells was elevated in the absence of METTL3 (Supplementary Fig. 4g, h). We further assessed the expression levels of genes that are associated with $T_{FH}$ cells, including *Tcf7*, *Cxcr5*, *Bcl6*, *Pdcd1*, and *Icos*. The expression levels of these genes were substantially decreased in *Mettl3*[fl/fl]*Cd4*-Cre SMARTA $T_{FH}$ cells compared with those of Ctrl cells (Fig. 3i). Correspondingly, the expressions of *Prdm1* and *Id2*, both well-known for promoting $T_H1$ differentiation[9,32], were much lower in Ctrl $T_{FH}$ cells than that of *Mettl3*[fl/fl]*Cd4*-Cre SMARTA cells (Fig. 3i). These results thus demonstrated that METTL3 is indispensable for early $T_{FH}$ commitment during acute viral infection.

**METTL3 orchestrates the transcriptional profiles of $T_{FH}$ cells.** We next investigated how METTL3 deficiency affects the transcriptional profiles of $T_{FH}$ cells. Considering CD4[+] T cells generally differentiate into $T_H1$ cells or $T_{FH}$ cells upon acute viral infection[33] and METTL3 deficiency affects CXCR5 expression, we hence used CD44 and SLAM, which is expressed at low level on $T_{FH}$ cells[9], to identify $T_H1$ and $T_{FH}$ cells in activated CD4[+] T cells. CD44[+]SLAM[hi] $T_H1$ cells and CD44[+]SLAM[lo] $T_{FH}$ cells were sorted from *Mettl3*[fl/fl]*Cd4*-Cre mice and their Ctrl littermates on day 8 post viral infection and subjected to RNA-seq analysis. Compared with those in wild-type cells, 515 upregulated genes and 252 downregulated genes in METTL3-deficient $T_{FH}$ cells (Fig. 4a) and 763 upregulated genes and 332 downregulated genes (Supplementary Fig. 5a) in METTL3-deficient $T_H1$ cells were identified, respectively (≥2-fold expression change, adjusted $P$ value <0.01). In particular, the upregulated genes in $T_{FH}$ cells were significantly enriched in the T-cell differentiation and defense response to virus; the downregulated genes in $T_{FH}$ cells were apparently enriched in T-cell proliferation and differentiation (Fig. 4b). Then we selected a $T_{FH}$ cell and a GC $T_{FH}$ cell signature gene set[11] for gene set enrichment analysis (GSEA). Both the $T_{FH}$ lineage gene set and the GC $T_{FH}$-associated gene set were enriched in wild-type $T_{FH}$ cells, but not in METTL3-deficient $T_{FH}$ cells (Fig. 4c). When applying GSEA to exhibit the signature genes of $T_H1$ (ref. [12]), $T_H2$ (ref. [34]), $T_H17$ (ref. [35]), and Treg[34] cells, we found that all of which were enriched in METTL3-deficient $T_{FH}$ cells (Supplementary Fig. 5b). Similarly, we also observed $T_{FH}$, $T_H2$, $T_H17$, and Treg-related gene sets were positively enriched in METTL3-deficient $T_H1$ cells (Supplementary Fig. 5c). These analyses indicated that loss of METTL3 leads to disordered gene profiles of both $T_{FH}$ and $T_H1$ transcription program.

We next validated expression changes of key $T_{FH}$ genes and found that *Cxcr5*, *Bcl6*, *Pdcd1*, and *Icos* were significantly lower in METTL3-null $T_{FH}$ cells than in wild-type cells (Fig. 4e). In addition, the expression of *Tcf7* was lower in METTL3-deficient $T_{FH}$ cells than in wild-type $T_{FH}$ cells (Fig. 4e). Among the genes with low expression, *Tcf7* was of interest because of its role in Tfh

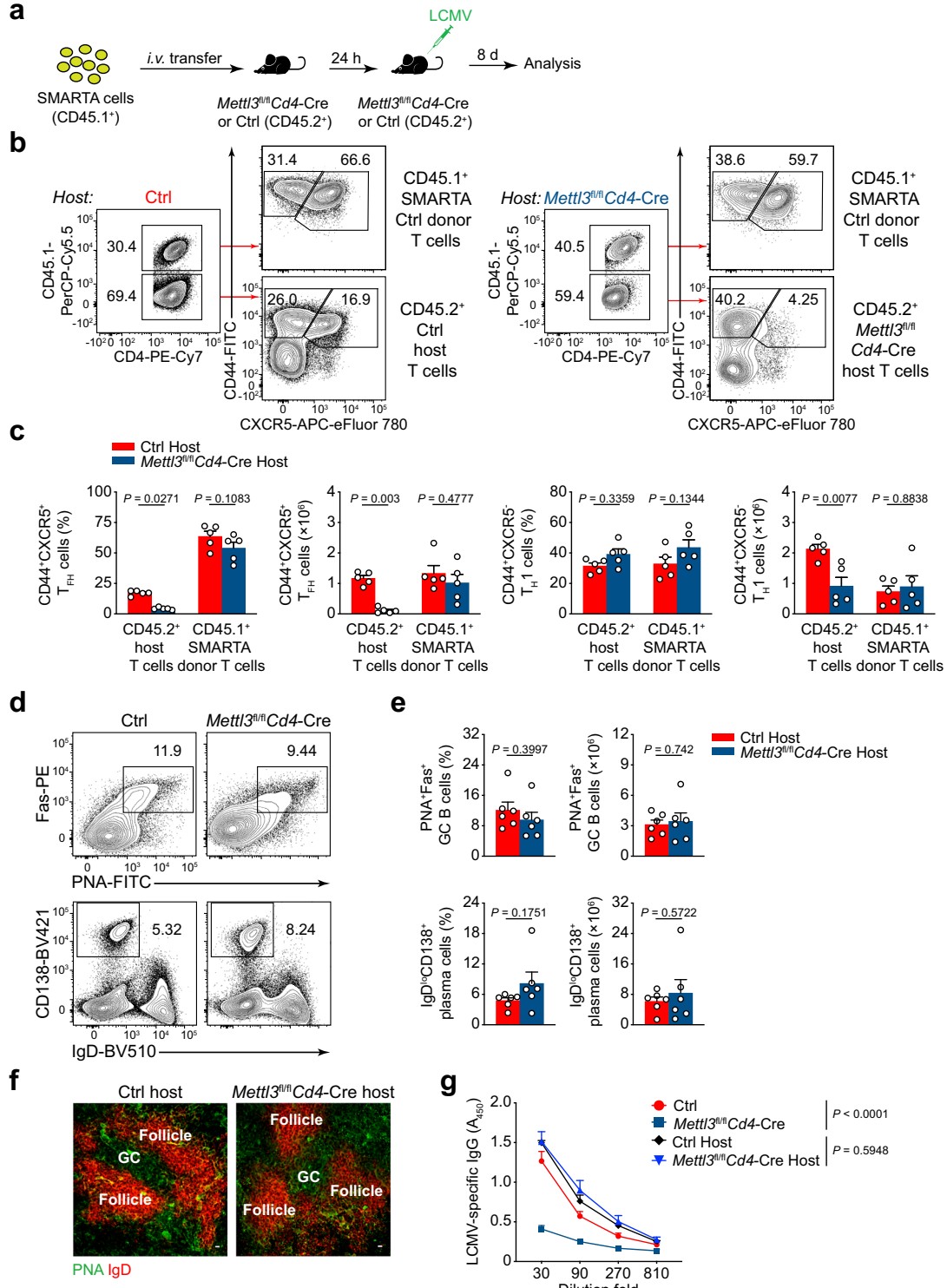

**Fig. 2 METTL3 intrinsically controls $T_{FH}$ differentiation and GC responses. a** Scheme of adoptive transfer model. $5 \times 10^6$ CD45.1+ SMARTA cells were adoptively transferred into CD45.2+ *Mettl3*fl/fl*Cd4*-Cre mice or Ctrl host mice, followed by LCMV-Armstrong infection within 24 h. **b, c** Flow cytometry analysis of CD45.1+ SMARTA or CD45.2+ host T cell-derived CD44+CXCR5+ $T_{FH}$ and CD44+CXCR5− $T_H$1 cells, gated on splenic CD4+ T cells from host mice on day 8 post viral infection. Frequency and numbers of CD44+CXCR5+ $T_{FH}$ cells and CD44+CXCR5− $T_H$1 cells are shown in **c** (*n* = 5 per group). **d, e** Flow cytometry analysis of splenic PNA+Fas+ GC B cells (top panel) and IgDloCD138+ plasma cells (bottom panel) from host mice on 8 *dpi*. Summary of the frequency and cell numbers of GC B cells and plasma cells are shown in **e** (*n* = 6 per group). **f** Immunofluorescent staining of spleens from Ctrl or *Mettl3*fl/fl*Cd4*-Cre host mice on 8 *dpi*. Green: PNA; Red: IgD; scale bar: 10 μm. **g** Analysis of LCMV-specific IgG concentration in serum on 8 *dpi* by ELISA (*n* = 5 for Ctrl group, *n* = 5 for *Mettl3*fl/fl*Cd4*-Cre group, *n* = 6 for Ctrl host group, *n* = 6 for *Mettl3*fl/fl*Cd4*-Cre host group). Data are representative of at least three independent experiments. Error bars indicate standard error of the mean. *P* value was calculated by unpaired two-tailed Student's *t* test (**e**) or two-way ANOVA coupled with multiple comparisons (**c, g**).

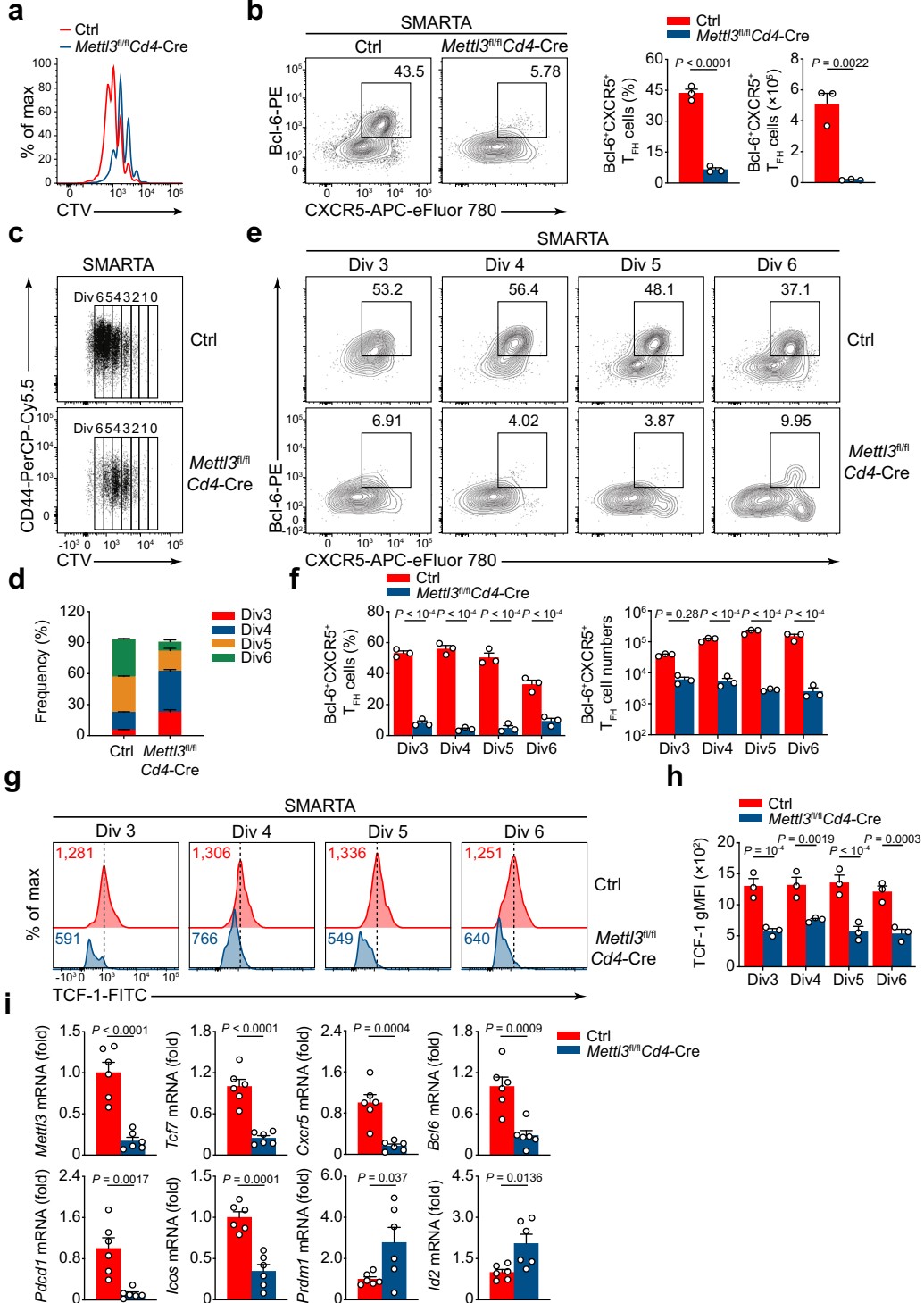

**Fig. 3 METTL3 is indispensable for the initiation of T_FH development. a** Flow cytometry analysis of cells from the wild-type recipient mice (CD45.1$^+$) given adoptive transfer of naive CTV-labeled Ctrl or *Mettl3*$^{fl/fl}$*Cd4*-Cre SMARTA cells, followed by LCMV-Armstrong infection and analysis 3 days later as CTV dilution by the transferred cells. **b** Flow cytometry analysis of Bcl-6$^+$CXCR5$^+$ T_FH cells gated on SMARTA CD4$^+$ T cells from recipient mice as in **a**. Summary of the frequency and cell numbers of T_FH cells are shown on the right (*n* = 3 per group). **c, d** Flow cytometry analysis of different cell divisions gated on SMARTA CD4$^+$ T cells from recipient mice as in **a**. The frequency of third to sixth divisions is summarized in **d** (*n* = 3 per group). **e, f** Contour plots display Bcl-6$^+$CXCR5$^+$ T_FH cells in indicated divisions, among SMARTA cells from recipient mice as in **a**. Summary of the frequency and cell numbers of T_FH cells in indicated divisions are shown in **f** (*n* = 3 per group). **g, h** Flow cytometry analysis of the expression of TCF-1 in CXCR5$^+$ cells in indicated divisions gated on SMARTA CD4$^+$ T cells from recipient mice as in **a**. Quantification of geometric mean fluorescence intensity (gMFI) of TCF-1 in indicated divisions is shown in **h** (*n* = 3 per group). **i** Quantitative RT-PCR analysis of mRNA abundance of T_FH cell-related genes in CD25$^-$CXCR5$^+$ T_FH cells from recipient mice as in **a**, relative expression was normalized to Ctrl cells (*n* = 6 per group). Data are representative of two independent experiments. Error bars indicate standard error of the mean. *P* value was calculated by unpaired two-tailed Student's *t* test (**b, i**) or two-way ANOVA coupled with multiple comparisons (**f, h**).

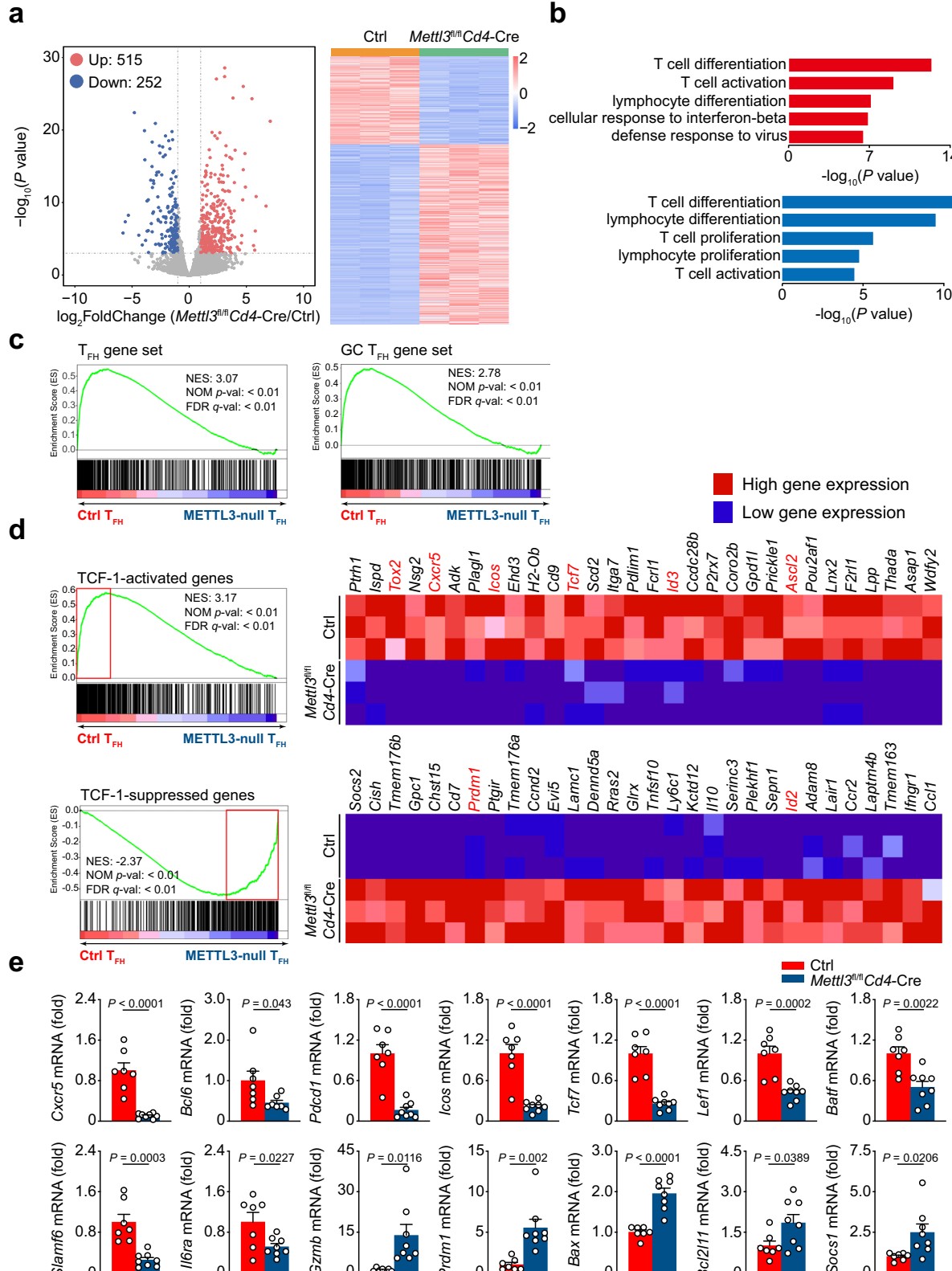

**Fig. 4 Alternation of transcriptional profiles in METTL3-null T_FH cells. a** Volcano map depicting genes upregulated (red) or downregulated (blue) 2-fold or more in T_FH cells on 8 *dpi*. Heatmap of differentially expressed genes is shown on the right. **b** Gene Ontology (GO) terms of the differentially expressed genes in *Mettl3*^fl/fl^*Cd4*-Cre T_FH cells compared with Ctrl T_FH cells (Upregulated: top panel; Downregulated: bottom panel). **c** Gene set enrichment analysis (GSEA) of T_FH and GC T_FH gene set in *Mettl3*^fl/fl^*Cd4*-Cre T_FH cells relative to expression in Ctrl T_FH cells as in **a**. **d** GSEA analysis of "TCF-1-activated genes in T_FH cells" and "TCF-1-suppressed genes in T_FH cells" (GSE65693) in *Mettl3*^fl/fl^*Cd4*-Cre T_FH cells relative to expression in Ctrl T_FH cells as in **a**. Heatmaps representation of top 30 ranking genes in the leading edge are shown, respectively. **e** Quantitative RT-PCR analysis of selected interested genes in **a**, relative expression was normalized to that in Ctrl T_FH cells (*n* = 7 for Ctrl group, *n* = 8 for *Mettl3*^fl/fl^*Cd4*-Cre group except *Bcl6* gene (*n* = 7 per group)). Data are from one experiment with triplicates (**a**–**d**) or pooled from two experiments (**e**). Error bars indicate standard error of the mean. *P* value was calculated by unpaired two-tailed Student's *t* test (**e**).

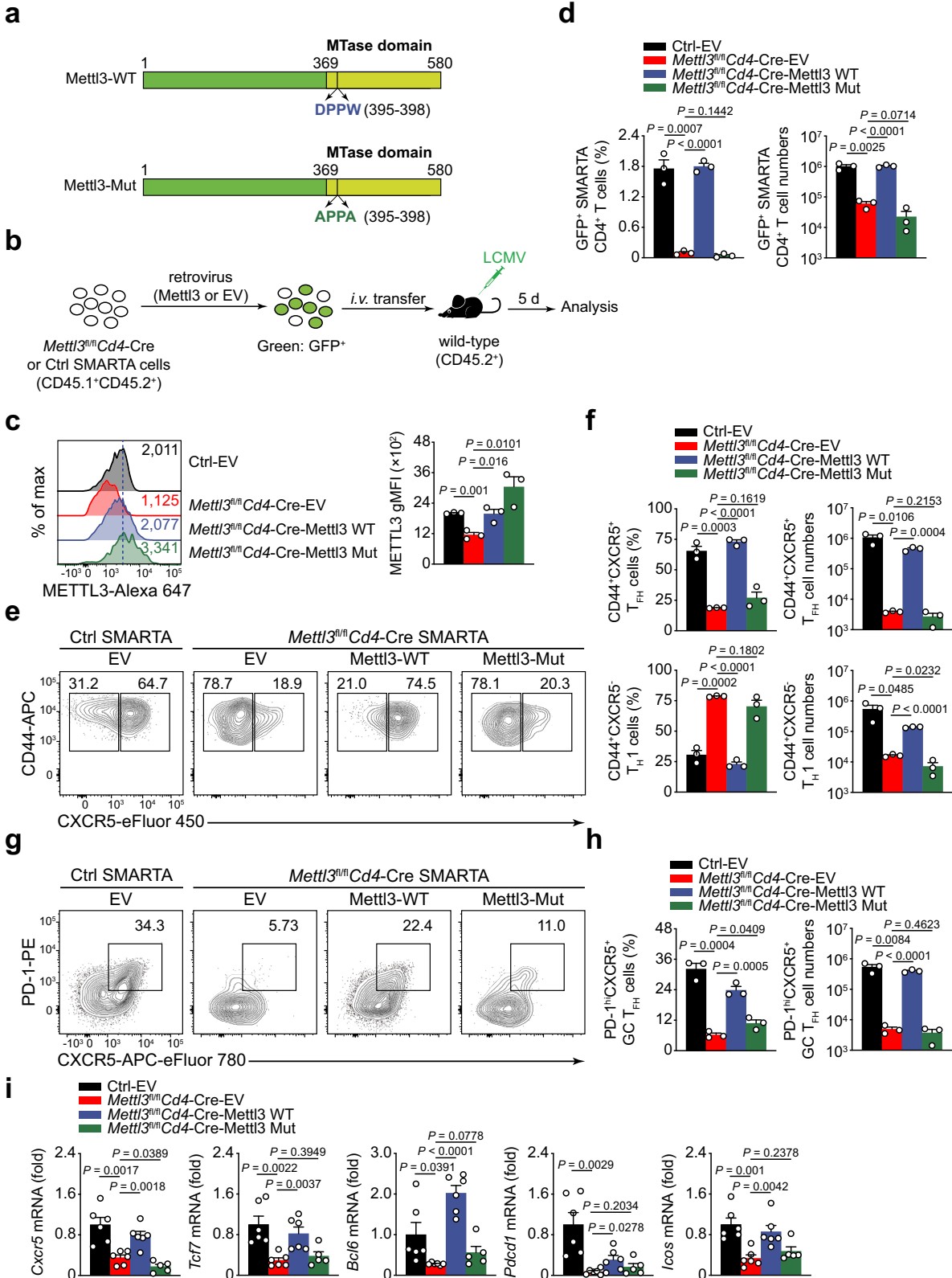

cell differentiation, so we also performed GSEA analysis of a gene set containing TCF-1-activated genes in $T_{FH}$ cells[13]. Interestingly, remarkable enrichment was exhibited in wild-type $T_{FH}$ cells, whereas the TCF-1-suppressed gene set[13] was notably enriched in METTL3-deficient cells, which suggested that METTL3 and TCF-1 share a common subset of target genes in

$T_{FH}$ cells (Fig. 4d). Meanwhile, the expression of *Lef1* mRNA was also decreased in METTL3-deficient $T_{FH}$ cells compared with wild-type cells (Fig. 4e). On the other hand, the expression of *Prdm1*, known as the antagonist of *Bcl6*, and *Gzmb* was higher in METTL3-null $T_{FH}$ cells than that in wild-type $T_{FH}$ cells (Fig. 4e). The expression levels of *Bcl2l11* and *Bax*, both known as

**Fig. 5 METTL3 promotes T$_{FH}$ differentiation in an m$^6$A catalytic activity-dependent manner. a** Graphic representation of the wild-type (Mettl3-WT) and catalytic domain dead (Mettl3-Mut; DPPW to APPA) METTL3 constructs. **b** Scheme of retrovirus transduction experiment. SMARTA cells were transduced with indicated structures by using a retrovirus transduction system. Then, the transduced cells were adoptively transferred into congenic CD45.2$^+$ wild-type mice followed by LCMV-Armstrong infection, and analyzed on day 5 post viral infection. **c** Flow cytometry analysis of METTL3 gMFI in GFP$^+$CD4$^+$ SMARTA cells from recipient mice. Quantitation of METTL3 gMFI is shown on the right ($n = 3$ per group). **d** Summary of the frequency and cell numbers of retrovirus-transduced SMARTA CD4$^+$ T cells in host mice ($n = 3$ per group). **e, f** Flow cytometry analysis of CD44$^+$CXCR5$^+$ T$_{FH}$ populations and CD44$^+$CXCR5$^-$ T$_H$1 subsets gated on SMARTA GFP$^+$CD4$^+$ T cells from host mice adoptively transferred with empty vector (EV), Mettl3-WT, or Mettl3-Mut retrovirus-introduced SMARTA cells. Summary of the frequency and cell numbers of T$_{FH}$ cells and T$_H$1 cells are shown in **f** ($n = 3$ per group). **g, h** Flow cytometry analysis of PD-1$^{hi}$CXCR5$^+$ GC T$_{FH}$ populations gated on SMARTA GFP$^+$CD4$^+$ T cells from host mice adoptively transferred with EV, Mettl3-WT, or Mettl3-Mut retrovirus-introduced SMARTA cells. Summary of the frequency and cell numbers of GC T$_{FH}$ cells are shown in **h** ($n = 3$ per group). **i** Quantitative RT-PCR analysis of mRNA abundance of T$_{FH}$ cell-related genes in CD44$^+$CXCR5$^+$ T$_{FH}$ cells ectopically expressed with EV, Mettl3-WT, or Mettl3-Mut, relative expression was normalized to Ctrl cells transduced with EV retrovirus ($n = 6$ for Ctrl EV, *Mettl3*$^{fl/fl}$*Cd4*-Cre-EV, and *Mettl3*$^{fl/fl}$*Cd4*-Cre-Mettl3 WT group; $n = 5$ for *Mettl3*$^{fl/fl}$*Cd4*-Cre-Mettl3 Mut group). Data are representative of two independent experiments. Error bars indicate standard error of the mean. *P* value was calculated by one-way ANOVA, followed by unpaired two-tailed Student's *t* test for indicated pairwise comparisons.

pro-apoptotic regulators[36,37], were significantly elevated in METTL3-null T$_{FH}$ cells compared with those in the wild-type T$_{FH}$ cells (Fig. 4e), which corresponded to accelerated apoptosis in *Mettl3*$^{fl/fl}$*Cd4*-Cre T$_{FH}$ cells (Supplementary Fig. 4g, h). In addition, the transcripts encoding other essential transcription factors, receptors and ligands for T$_{FH}$ differentiation, including *Batf*, *Slamf6*, and *Il6ra*, were downregulated in *Mettl3*$^{fl/fl}$*Cd4*-Cre T$_{FH}$ cells compared with those in wild-type cells (Fig. 4e). Besides, we observed elevated expression of *Socs1* mRNA, a known target of METTL3[25,26], in *Mettl3*$^{fl/fl}$*Cd4*-Cre T$_{FH}$ cells (Fig. 4e). Collectively, these results suggested that METTL3 regulates T$_{FH}$ differentiation program by activating T$_{FH}$ program but suppressing T$_H$1 lineage-associated genes.

**METTL3 promotes T$_{FH}$ differentiation in an m$^6$A catalytic activity-dependent manner.** m$^6$A methylation is catalyzed by a multicomponent methyltransferase complex, while METTL3 functions as the predominant catalytic subunit[15,16]. Next, we analyzed whether METTL3-mediated T$_{FH}$ differentiation is dependent on its m$^6$A methyltransferase activity. To achieve this goal, we generated wild-type METTL3 (Mettl3-WT) and catalytic domain mutant METTL3 (D395A and W398A; Mettl3-Mut) constructs[15] (Fig. 5a). These METTL3 proteins were then expressed in *Mettl3*$^{fl/fl}$*Cd4*-Cre SMARTA CD4$^+$ T cells with a retrovirus transduction system (Fig. 5b). On day 5 post infection, forced expression of Mettl3-WT and Mettl3-Mut both resulted in elevated METTL3 expression in *Mettl3*$^{fl/fl}$ *Cd4*-Cre SMARTA cells compared with the empty-vector (EV) retrovirus (Fig. 5c). EV retrovirus-infected *Mettl3*$^{fl/fl}$*Cd4*-Cre SMARTA CD4$^+$ T cells exhibited reduced expansion than EV retrovirus-infected Ctrl cells, which could be restored by overexpression of Mettl3-WT but not Mettl3-Mut (Fig. 5d). Moreover, EV retrovirus-infected *Mettl3*$^{fl/fl}$*Cd4*-Cre SMARTA CD4$^+$ T cells remained defective in the generation of CD44$^+$CXCR5$^+$ T$_{FH}$ cells, while Mettl3-WT but not Mettl3-Mut retrovirus promoted differentiation of *Mettl3*$^{fl/fl}$*Cd4*-Cre SMARTA CD4$^+$ T cells into a T$_{FH}$ fate (Fig. 5e). In addition, forced expression of Mettl3-WT could restore both the cell numbers of *Mettl3*$^{fl/fl}$*Cd4*-Cre T$_{FH}$ cells and T$_H$1 cells (Fig. 5f). Accordingly, ectopic expression of Mettl3-WT could largely rectify defective PD-1$^{hi}$CXCR5$^+$ GC T$_{FH}$ cells in METTL3-deficient cells (Fig. 5g, h). We also observed that the expression of T$_{FH}$ cell-related genes, including *Cxcr5*, *Tcf7*, *Bcl6*, *Pdcd1*, and *Icos*, were restored in *Mettl3*$^{fl/fl}$*Cd4*-Cre cells with inducing Mettl3-WT expression (Fig. 5i). The data collectively indicated that the m$^6$A-catalytic activity of METTL3 is necessary for T$_{FH}$ differentiation.

**m$^6$A modifies *Tcf7* mRNA in 3′ UTR to control its stability.** To examine how m$^6$A methylation modulates T$_{FH}$ transcriptional program, we performed m$^6$A-miCLIP-SMARTer-seq to map global m$^6$A landscape in Ctrl or *Mettl3*$^{fl/fl}$*Cd4*-Cre SMARTA CD4$^+$ T cells primed in vivo. Bioinformatic analysis revealed that m$^6$A peaks were significantly abundant in 3′ UTR and near the stop codon of mRNAs (Fig. 6a, b). Approximately 45.3% methylated mRNAs contained three or more peaks (Fig. 6c). Biological duplicates of m$^6$A-miCLIP-SMARTer-seq yielded about 4939 transcripts (called 'm$^6$A-modified transcripts' hereafter; Fig. 6d). By stratifying m$^6$A-modified transcripts with differentially expressed genes, we found that 208 differentially expressed transcripts were potentially regulated by m$^6$A methylation (Fig. 6d). GO term analysis showed these transcripts were enriched for functions related to defense response to the virus, T-cell activation and differentiation (Fig. 6e). Among these potential m$^6$A methylated target genes, a set of T$_{FH}$ cell-relevant genes were directly marked by m$^6$A, such as *Cxcr5*, *Icos*, *Il6ra*, and *Il6st* (Supplementary Fig. 6a). Interestingly, the 3′ UTR of *Tcf7* mRNA had highly enriched m$^6$A peaks, whereas *Bcl6* and *Prdm1* mRNAs were not tagged by m$^6$A (Fig. 6f and Supplementary Fig. 6a). Accordingly, RNA immunoprecipitation (RIP) assay suggested METTL3 directly binds to *Tcf7* mRNA (Fig. 6g). The m$^6$A-RIP-qPCR results further indicated *Tcf7* mRNA was tagged by m$^6$A methylation and the m$^6$A levels on *Tcf7* mRNA were substantially decreased in METTL3-deficient cells (Fig. 6f, h). These data jointly indicated *Tcf7* is a bona fide m$^6$A target. Accordingly, both the *Tcf7* mRNA and the TCF-1 protein were significantly decreased in METTL3-deficient T$_{FH}$ cells compared with those in wild-type T$_{FH}$ cells (Fig. 4e and Fig. 6i). To further validate METTL3 regulation of *Tcf7* expression in an m$^6$A-dependent manner, we identified a high-confidence m$^6$A site (GGA$_{1011}$CT, a conserved m$^6$A methylation motif) in the *Tcf7* 3′ UTR region (Fig. 6f). We generated a minigene vector placing the *Tcf7* m$^6$A site to the 3′ end of luciferase reporter cDNA. Then the 'GGACT' consensus motif in the *Tcf7* m$^6$A site was mutated into 'GGTCT' to abrogate m$^6$A modification (Fig. 6j). Wild-type *Tcf7* 3′ UTR exhibited increased luciferase activity compared to that of the empty vector (pGL4.23) in the presence of overexpressed METTL3; notably, mutation of *Tcf7* 3′ UTR almost completely compromised the increase (Fig. 6k). These results revealed that METTL3 regulates *Tcf7* mRNA levels via m$^6$A modifications in the 3′ UTR region (m$^6$A$_{1011}$). Given m$^6$A modification regulates gene expression by multiple approaches, including mRNA splicing, stability, and translation[17,38], alternative splicing assay was applied and exhibited no significant difference in the alternative splicing of *Tcf7* mRNA between Ctrl and METTL3-deficient T$_{FH}$ cells (Supplementary Fig. 6b). We then performed RNA decay

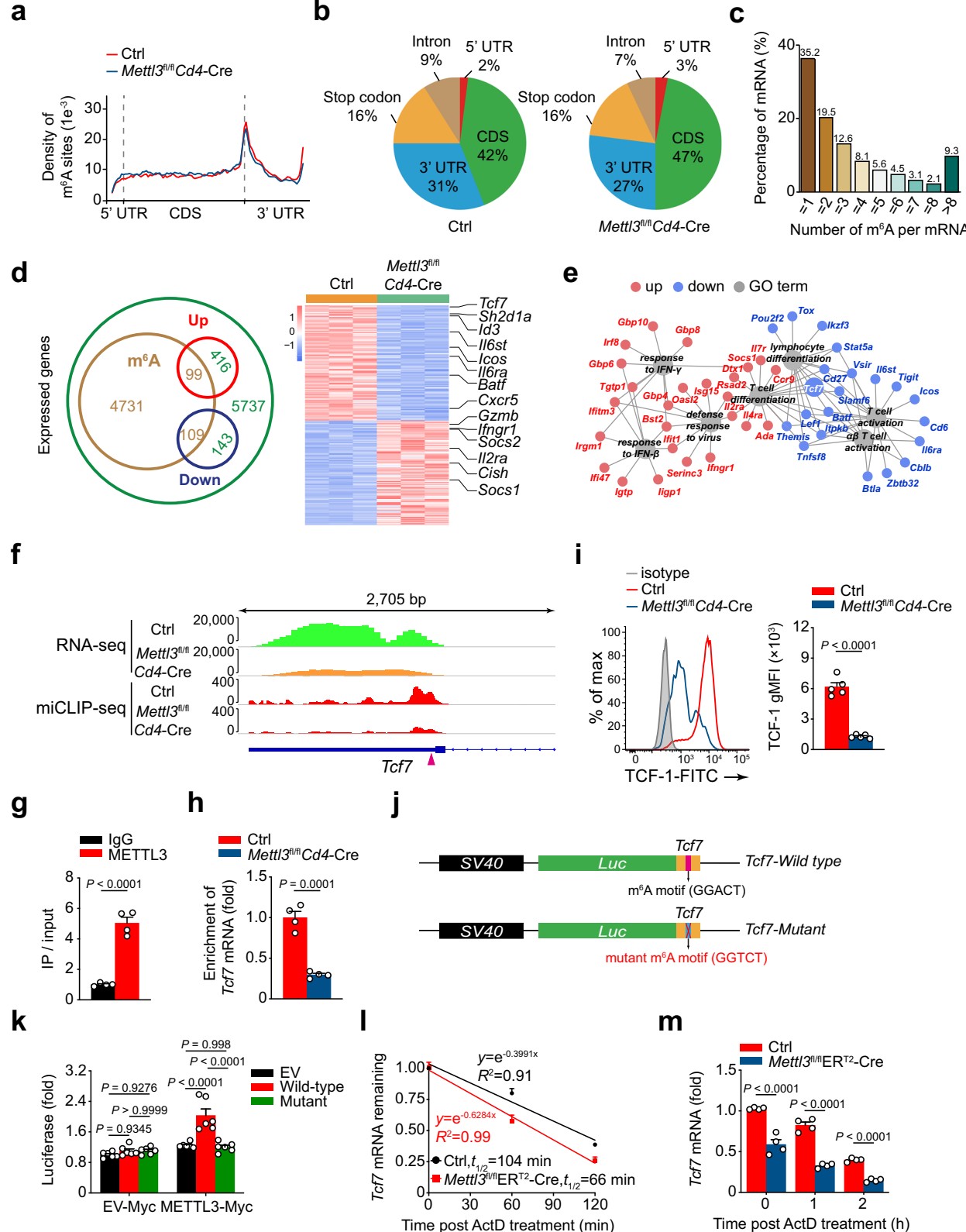

assay via actinomycin D (ActD) treatment to detect the stability of *Tcf7* mRNA, and found that *Tcf7* mRNA exhibited substantially accelerated decline in METTL3-deficient cells compared with Ctrl cells at checkpoints after treatment (Fig. 6l, m). Taken together, these data suggested that METTL3 enhances *Tcf7* mRNA stability via catalyzing m6A methylation at its 3′ UTR, to ensure TCF-1 expression in promoting $T_{FH}$ differentiation.

**Forced expression of TCF-1 restores defective $T_{FH}$ differentiation in METTL3-deficient mice.** To determine the functional link between METTL3 and *Tcf7* transcript stability in $T_{FH}$ differentiation, we next examined the impact of ectopic expression of TCF-1 in METTL3-deficient cells. Upon transducing in vivo primed SMARTA CD4+ T cells with TCF-1 (full-length CDS of P45 isoform without 3′ untranslated region[39]) retrovirus

**Fig. 6 m⁶A modifies *Tcf7* mRNA to control its stability. a** Metagene profiles of m⁶A site distribution along a normalized transcript containing three rescaled non-overlapping segments: 5′ UTR, CDS, and 3′ UTR in Ctrl and *Mettl3*<sup>fl/fl</sup>*Cd4*-Cre SMARTA CD4⁺ T cells. **b** Pie chart showing the distribution of m⁶A sites in five regions of Ctrl and *Mettl3*<sup>fl/fl</sup>*Cd4*-Cre SMARTA CD4⁺ T cells. **c** Bar chart depicting percentage of mRNAs with different internal m⁶A abundance. **d** Venn diagram showing the overlapping differentially expressed genes from RNA-seq and m⁶A-modified transcripts from m⁶A-miCLIP-SMARTer-seq. Heatmap of 208 differentially expressed genes with m⁶A modification is shown on the right. **e** Representative GO terms of the biological process categories enriched in differentially expressed transcripts with m⁶A peaks (Red: upregulated genes; blue: downregulated genes; gray: GO terms). **f** Integrative Genomics Viewer (IGV) tracks displaying RNA-seq (top panel) and m⁶A-miCLIP-SMARTer-seq (bottom panel) reads distribution of *Tcf7* gene. The high-confidence m⁶A site is marked as a triangle. **g** RIP-qPCR analysis showing enrichment of METTL3 on *Tcf7* mRNA in SMARTA CD4⁺ T cells. The *Tcf7* mRNA enrichment is presented as IP/input and normalized to IgG group ($n = 4$ per group). **h** m⁶A-RIP-qPCR analysis of m⁶A enrichment on *Tcf7* mRNA of Ctrl and *Mettl3*<sup>fl/fl</sup>*Cd4*-Cre SMARTA CD4⁺ T cells ($n = 4$ per group). **i** Flow cytometry analysis of expression level of TCF-1 on T$_{FH}$ cells on 8 dpi. Quantification of gMFI of TCF-1 is shown on the right ($n = 5$ per group). **j** Constructions of plasmids with wild-type or m⁶A site mutant in *Tcf7* 3′ UTR. **k** Luciferase reporter assay. Results were normalized to the luciferase activity of cells co-transfected with the pGL4.23 empty plasmid and EV-Myc plasmid ($n = 6$ per group). **l** RNA decay assay. The half-live of *Tcf7* mRNA was detected by quantitative RT-PCR. The remaining mRNAs were normalized to $t = 0$ ($n = 4$ per group). **m** Quantitative RT-PCR analysis of *Tcf7* mRNA abundance as in **l**, relative expression was normalized to $t = 0$ of Ctrl cells ($n = 4$ per group). Data are from one experiment with duplicate (**a–e**) or representative of at least three independent experiments (**i–h**, **k–m**). Error bars indicate standard error of the mean. *P* value was calculated by unpaired two-tailed Student's *t* test (**g–i**) or two-way ANOVA coupled with multiple comparisons (**k**, **m**).

(Fig. 7a), on day 8 post LCMV-Armstrong infection, we observed a significant increase in TCF-1 expression in METTL3-deficient SMARTA cells (Fig. 7b). We then analyzed T$_{FH}$ populations from transduced SMARTA CD4⁺ T cells in recipient mice. Compared with Ctrl SMARTA CD4⁺ T cells infected with EV retrovirus, the EV-infected *Mettl3*<sup>fl/fl</sup>*Cd4*-Cre SMARTA CD4⁺ T cells exhibited defects in CD44⁺CXCR5⁺ T$_{FH}$ differentiation; whereas TCF-1 retrovirus could largely rectify the ability of *Mettl3*<sup>fl/fl</sup>*Cd4*-Cre SMARTA CD4⁺ T cells to differentiate into T$_{FH}$ cells (Fig. 7c). In addition, TCF-1 overexpression also elevated the cell numbers of *Mettl3*<sup>fl/fl</sup>*Cd4*-Cre T$_{FH}$ cells, but not T$_H$1 cells (Fig. 7d). Consistently, PD-1<sup>hi</sup>CXCR5⁺ GC T$_{FH}$ cells could also be rescued with TCF-1 overexpression (Fig. 7e, f). Meanwhile, forced expression of TCF-1 could largely restore the expression levels of TCF-1, CXCR5, PD-1, ICOS, and Bcl-6 on *Mettl3*<sup>fl/fl</sup>*Cd4*-Cre T$_{FH}$ cells (Fig. 7g, h). These data collectively demonstrated that METTL3 stabilizes *Tcf7* mRNA expression to promote T$_{FH}$ differentiation.

## Discussion

$N^6$-methyladenosine (m⁶A) accounts for the prevalent mRNA modifications and has recently emerged as a critical epitranscriptomic regulator to affect the translation and stability of the modified transcripts. Here, we showed that ablation of METTL3 leads to substantial defects in both T$_{FH}$ and T$_H$1 differentiation upon LCMV-Armstrong and KLH challenges. Based on our results, a more severe phenotype was exhibited in T$_{FH}$ cells than that of T$_H$1 cells from METTL3-deficient mice, which strongly supports the notion that the METTL3 acts as an intrinsic modulator of T$_{FH}$ cells.

During acute viral infection, bifurcation of effector CD4⁺ T cells into T$_{FH}$ cells or T$_H$1 cells can be observed as early as the second to third division[40]. A recent report demonstrated that METTL3-null CD4⁺ T cells remained naive state and exhibited defective proliferation when tested both in vivo and in vitro[25]. Using an adoptive transfer model, we found that METTL3-deficient CD4⁺ T cells showed a delayed differentiation and a slower proliferation upon acute viral infection, although they were activated normally as shown by expression of the surface markers via both in vivo and in vitro assay. Meanwhile, at 72 h post infection, the activated CD4⁺ T cells were mostly in the fifth and sixth divisions, whereas METTL3-deficient CD4⁺ T cells were dominant in third and fourth divisions. These METTL3-deficient CD4⁺ T cells also showed defects in differentiation into T$_{FH}$ cells and exhibited elevated apoptosis. Further, METTL3

deficiency also resulted in a decreased expression of T$_{FH}$-defining genes in these nascent T$_{FH}$ cells. Strikingly, the abundances of *Tcf7* and *Bcl6* mRNAs, both essential for the early commitment of T$_{FH}$ cells[41], were profoundly decreased in early METTL3-null T$_{FH}$ cells. Correspondingly, the protein levels of TCF-1 and Bcl-6 were also notably reduced. These findings revealed that METTL3 is required for early T$_{FH}$ commitment, proliferation, and survival by maintaining the key T$_{FH}$ gene expression.

As an RNA binding protein, METTL3 is the essential catalytic component of the conserved heterodimeric m⁶A writer complex[15]. Does METTL3-mediated T$_{FH}$ differentiation also directly depend on its m⁶A catalytic activity? Our results showed that forced expression of METTL3 with a mutated catalytic domain failed to rectify the defects in T$_{FH}$ differentiation of *Mettl3*<sup>fl/fl</sup>*Cd4*-Cre CD4⁺ T cells, while wild-type METTL3 did. We thus concluded that METTL3 promotes T$_{FH}$ generation in an m⁶A catalytic activity-dependent manner. Giving m⁶A methylation is essential for gene expression regulation[42], we hence focused on the methylation status of those differentially expressed genes. The expression patterns of a series of T$_{FH}$-defining genes were altered at both early and mature stages, including *Tcf7*, *Cxcr5*, *Bcl6*, *Pdcd1*, and *Icos*. Through m⁶A-miCLIP-SMARTer-seq, we found *Tcf7* mRNA was tagged by m⁶A in the 3′ UTR, whereas the other key T$_{FH}$ cell regulator *Bcl6* mRNA was not modified. Further study indicated that METTL3-mediated m⁶A modification regulates the stability of *Tcf7* mRNA, ultimately maintains TCF-1 level. Moreover, forced expression of TCF-1 could restore the T$_{FH}$ differentiation in the absence of METTL3. These data collectively suggested that METTL3 directly modulates the *Tcf7* expression level in an m⁶A-dependent manner to promote T$_{FH}$ differentiation.

The functional outcome of m⁶A modification is determined by an expanding list of m⁶A reader proteins in a cell type and cellular context-dependent fashion[14,43]. Although early evidence demonstrated that m⁶A deposition destabilizes mRNAs, resulting in their faster decay, a recent study has reported that the insulin-like growth factor 2 mRNA binding proteins (IGF2BP1-3), as a class of distinct m⁶A readers, promote m⁶A-modified mRNA stability and translation[20]. Similarly, a more recent study reported Prrc2a as a novel m⁶A reader that regulates oligodendrocyte specification and myelination by stabilizing target mRNA[44]. Our data also indicated m⁶A modification in the 3′ UTR of *Tcf7* enhances its mRNA stability to promote T$_{FH}$ differentiation. However, the currently known IGF2BPs family readers are expressed at extremely low levels to detect in CD4⁺ T cells (Supplementary Fig. 8), implying the possibility of other unidentified proteins might be responsible

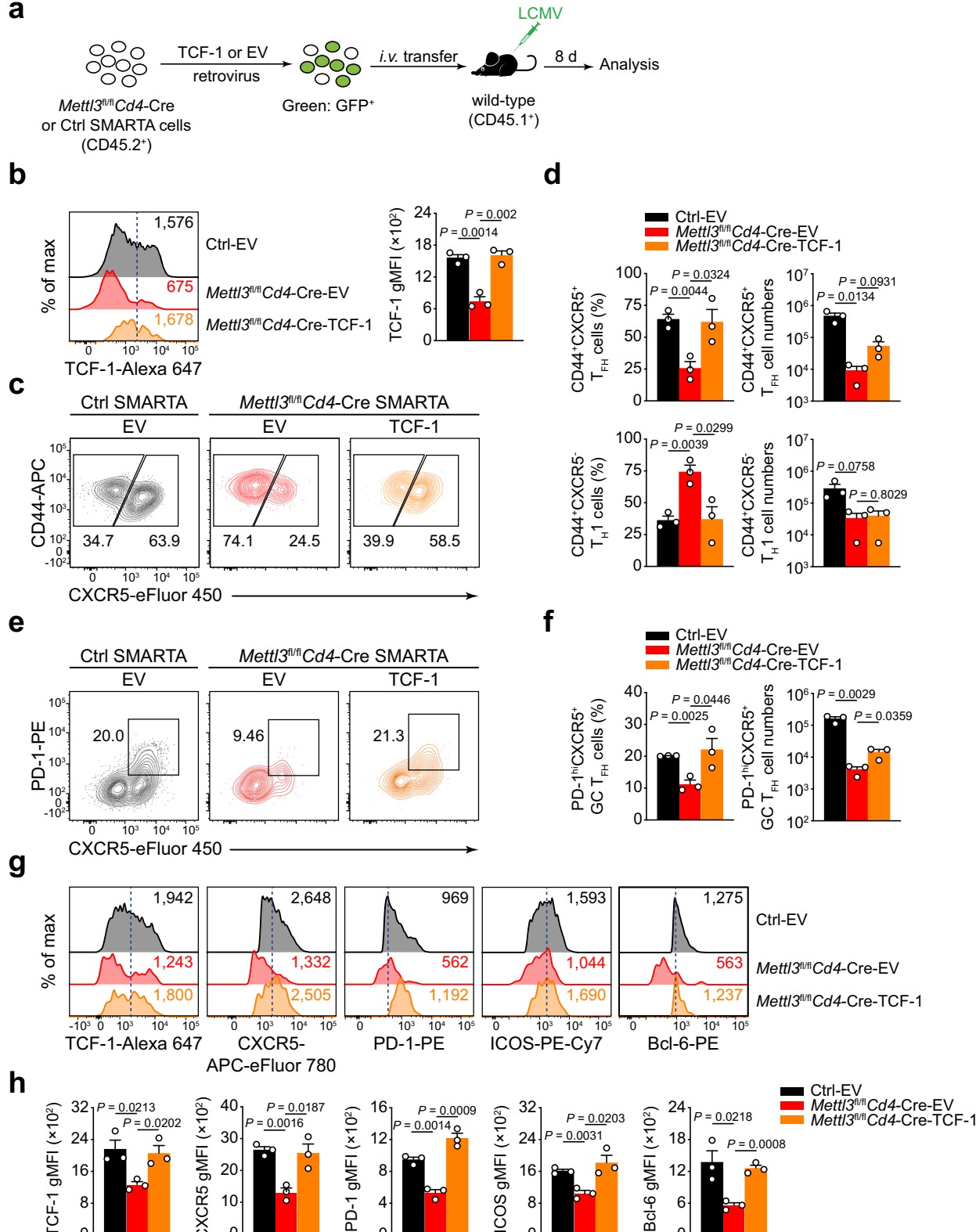

for deciphering the $m^6A$-modified transcripts in $T_{FH}$ cells. Therefore, it will be of great interest to identify new RNA-binding proteins virtually involved in $T_{FH}$ cells via post-transcriptional networks in future studies.

In METTL3-deficient cells, destabilization of *Tcf7* mRNA impacted $T_{FH}$ cells at least in two major aspects. The first aspect is

that loss of TCF-1 expression results in massive apoptosis in T cell[45], which is coordinated with the decreased $T_{FH}$ cell numbers in the absence of METTL3. The other point lies in that TCF-1 directly regulates *Bcl6* mRNA expression[11–13], and two of them thus collectively affect $T_{FH}$ differentiation. This post-transcriptional regulation of *Tcf7* mRNA represents a distinct

**Fig. 7 Enhanced TCF-1 expression rectifies defective T$_{FH}$ differentiation in METTL3-null cell. a** Scheme of TCF-1 rescue experiment. SMARTA cells were transduced with TCF-1 structure (full-length CDS of P45 isoform without 3′ UTR region) by using a retrovirus transduction system. Then, the transduced cells were adoptively transferred into congenic CD45.1$^+$ wild-type mice followed by LCMV-Armstrong infection, and analyzed on day 8 post viral infection. **b** Flow cytometry analysis of TCF-1 gMFI in SMARTA GFP$^+$CD4$^+$ cells from recipient mice on 8 *dpi*. Quantitation of TCF-1 gMFI is shown on the right ($n = 3$ per group). **c, d** Flow cytometry analysis of CD44$^+$CXCR5$^+$ T$_{FH}$ populations and CD44$^+$CXCR5$^-$ T$_H$1 subsets gated on SMARTA GFP$^+$CD4$^+$ T cells from different host mice adoptively transferred with empty vector (EV) or TCF-1 retrovirus-introduced SMARTA cells at 8 days post infection. Summary of the frequency and cell numbers of T$_{FH}$ cells and T$_H$1 cells are shown in **d** ($n = 3$ per group). **e, f** Flow cytometry analysis of PD-1$^{hi}$CXCR5$^+$ GC T$_{FH}$ populations gated on SMARTA GFP$^+$CD4$^+$ T cells from different host mice adoptively transferred with EV or TCF-1 retrovirus-introduced SMARTA cells. Summary of the frequency and cell numbers of GC T$_{FH}$ cells are shown in **f** ($n = 3$ per group). **g, h** Flow cytometry analysis of gMFIs of TCF-1, CXCR5, PD-1, ICOS, and Bcl-6 on CD44$^+$CXCR5$^+$ T$_{FH}$ cells transduced with EV or TCF-1 retrovirus. Quantification of the gMFIs is shown in **h** ($n = 3$ per group). Data are representative of three independent experiments. Error bars indicate standard error of the mean. *P* value was calculated by one-way ANOVA, followed by unpaired two-tailed Student's *t* test for indicated pairwise comparisons.

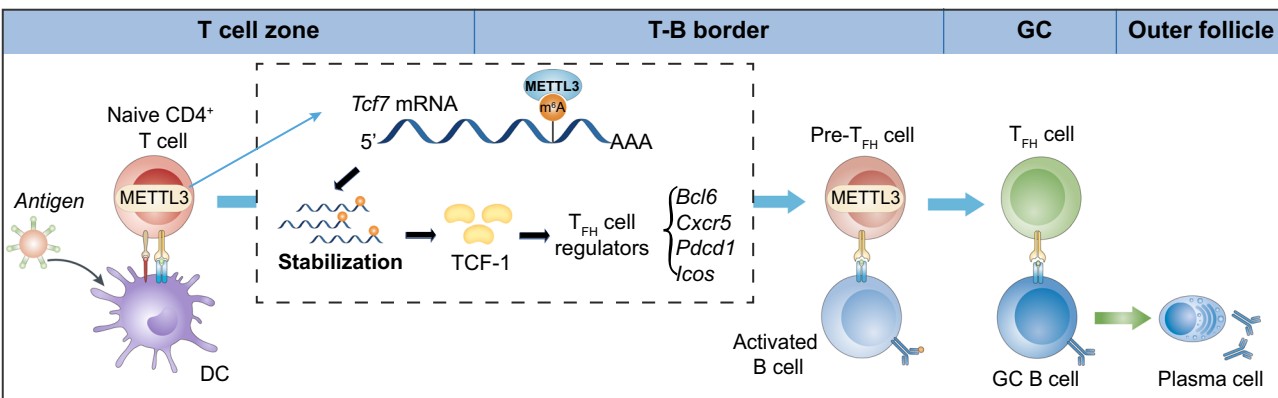

**Fig. 8 Proposed model for m$^6$A modification in promoting T$_{FH}$ differentiation.** During acute infection, METTL3-sufficient CD4$^+$ T cells were activated. With m$^6$A machinery, *Tcf7* mRNA was m$^6$A modified and stabilized, allowing normal production of TCF-1 protein. TCF-1 in turn regulates expressions of T$_{FH}$ cell regulators, which ultimately program T$_{FH}$ commitment, proliferation, survival, and functional maturation.

mechanism that drives T$_{FH}$ differentiation. It should be noted that other post-transcriptional regulators such as RNA-binding proteins and miRNAs also contribute to modulating T$_{FH}$ differentiation program[46–48]. Hence, exploring T$_{FH}$ fate determination on the layer of post-transcriptional level might be a fruitful effort in future investigations.

A recent study referred that induced GAPDH protein by VHL deficiency reduced *Icos* expression through METTL3/METTL14-catalyzed m$^6$A modification on *Icos* mRNA, implying that elevated m$^6$A modification on *Icos* mRNA in VHL-deficient cells decreases *Icos* expression which is associated with attenuated T$_{FH}$ differentiation[49]. By analyzing high-throughput data, we also observed m$^6$A modification in the 3′ UTR of *Icos* mRNA, and the m$^6$A level on *Icos* mRNA was decreased in the absence of METTL3. However, we found both the mRNA and protein level of ICOS were blunted in METTL3-deficient T$_{FH}$ cells, indicating that loss of m$^6$A modification impairs *Icos* expression. In addition, Zhu et al. reported that knockdown of METTL3 expression with short hairpin RNA (shRNA) in CD4$^+$ T cells could promote T$_{FH}$ differentiation[49], which differs in phenotypes from our genetic knockdown mice model. The varies may be contributed by the distinct experimental approaches and the divergent viewpoints from two studies also reflect the complex regulatory mechanism of m$^6$A modification, which needs to be further disclosed.

In summary, our study uncovers a critical role of METTL3-dependent m$^6$A methylation in directing T$_{FH}$ lineage differentiation. Conditional ablation of m$^6$A 'writer' METTL3 in CD4$^+$ T cells intrinsically impaired the T$_{FH}$ differentiation, proliferation, and survival. Consequently, the GC reactions were significantly compromised in METTL3-deficient mice in response to acute viral infection. Our data indicated that METTL3 directs

m$^6$A modification in 3′ UTR of *Tcf7* mRNA to stabilize the transcript and hence sustain TCF-1 protein expression (Fig. 8). Thus, m$^6$A functions as an important modulator of the METTL3-TCF-1 axis to initiate and secure the differentiation of T$_{FH}$ cells post-transcriptionally.

## Methods

**Mice.** *Mettl3*$^{fl/fl}$ mice were kindly provided by Drs. Qi Zhou and Wei Li (Institute of Zoology, Chinese Academy of Sciences). SMARTA mice[50] (expressing MHC II I-A$^b$-restricted TCR specific for LCMV glycoprotein amino acids 66–77) were generously provided by Dr. Rafi Ahmed (Emory University). *Cd4*-Cre, ER$^{T2}$-Cre, and C57BL/6J (CD45.2 and CD45.1) mice were purchased from the Jackson Laboratory. All mouse strains used in this study are on a fully C57BL/6J background. All mice were kept in group housing (3–5 mice per cage) in a specific pathogen-free facility with controlled environmental conditions of humidity (50 ± 10%), lighting (a 12-h light/dark cycle), and temperature (21 ± 1 °C) at China Agricultural University. All animal experiments were performed in accordance with the protocol of the Institutional Animal Care and Use Committee of China Agricultural University.

**LCMV infection.** LCMV-Armstrong strain was grown in BHK-21 cells and titers were determined as described before[51]. *Mettl3*$^{fl/fl}$*Cd4*-Cre mice and their wild-type littermates were intraperitoneally infected with $2 \times 10^5$ plaque-forming units (pfu) LCMV-Armstrong strain. In adoptive transfer experiments, recipient mice were infected 1 day after cell transfer. For bone marrow chimeric mice, LCMV infection was performed after 8 weeks reconstitution.

**Immunization.** *Mettl3*$^{fl/fl}$*Cd4*-Cre mice and their wild-type littermates were intraperitoneally immunized with 100 µg of KLH (Sigma-Aldrich) emulsified in CFA (Sigma-Aldrich). Eight days later, splenic T cells were analyzed.

**Flow cytometry and antibodies.** Single-cell suspensions of spleens were used for flow cytometry analysis or cell sorting. Surface staining was performed in PBS containing 1% FBS. The antibodies and reagents used for flow cytometry staining are listed as: anti-CD19 (1D3; 1:100), anti-CD25 (PC61.5; 1:100), anti-CD4 (RM4-5; 1:100), anti-CD44 (IM7; 1:100), anti-CD45.1 (A20; 1:100), anti-CD45.2

(104; 1:100), anti-CD62L (MEL-14; 1:100), anti-CD69 (H1.2F3; 1:100), anti-CD8a (53-6.7; 1:100), anti-B220 (RA3-6B2; 1:100), anti-GITR (DTA-1; 1:100), anti-GL7 (GL7; 1:100), anti-PD-1 (J43; 1:100), anti-TCR Vα2 (B20.1; 1:100) (from Thermo Fisher Scientific); anti-CD138 (281-2; 1:100), anti-Fas (Jo2; 1:100) (from BD Biosciences); anti-SLAM (TC15-12F12.2; 1:100), anti-ICOS (C398.4A; 1:100) (from BioLegend), and peanut agglutinin (PNA; Cat. no. FL-1071; 1:500; Vector laboratories). CXCR5 staining was performed with a three steps staining protocol as described before[52]. Briefly, single-cell suspensions were first stained with purified anti-CXCR5 (2G8; 1:100; BD Biosciences) for 1 h, followed by biotin-conjugated goat anti-rat IgG (Cat. no. 111-066-144; 1:1,000; Jackson ImmunoResearch) for 30 min, and then by APC-eFluor 780-, or eFluor 450-labeled streptavidin (1:500; Thermo Fisher Scientific) at 4 °C for 30 min in PBS supplemented with 2% normal mouse serum (Cat. no. 015-000-120; Jackson ImmunoResearch), 2% FCS, and 0.5% BSA. For detection of cytokines, the splenocytes from KLH-immunized mice were cultured in vitro for 5 h in 2 µg/mL of PMA (Cat. no. P8139; Sigma-Aldrich), 2 µg/mL of Ionomycin (Cat. no. I0634; Sigma-Aldrich), GolgiStop (BD Biosciences), and GolgiPlug (BD Biosciences). Intracellular staining for cytokines was performed with monoclonal antibody against to IL-4 (11B11; 1:100; Thermo Fisher Scientific) and IL-17a (TC11-18H10; 1:100; BD Biosciences), using the Fixation/Permeabilization buffer kit (BD Biosciences). For intracellular staining of Bcl-6 (K112-91; 1:20; BD Biosciences), Foxp3 (FJK-16s; 1:100; Thermo Fisher Scientific), GATA3 (TWAJ; 1:100; Thermo Fisher Scientific), RORγt (AFKJS-9; 1:100; Thermo Fisher Scientific), TCF-1 (C63D9; 1:100; Cell Signaling Technology), and METTL3 (Cat. no. ab195352; 1:100; Abcam), Foxp3/Transcription Factor Staining Buffer Set (Thermo Fisher Scientific) was used following the manufacturer's instructions. Active Caspase-3 was detected using the CaspGLOW™ Fluorescein Active Caspase-3 Staining Kit (Cat. no. 88-7004-42; 1:200; Thermo Fisher Scientific). All data were collected on a FACSVerse (BD Biosciences) with FACSSuite software (v1.0.5) or an LSRFortessa (BD Biosciences) with FACSDiva software (v8.0.2) and were analyzed with FlowJo software (v10; Treestar). The gating strategies for flow cytometry data analysis are illustrated in Supplementary Fig. 7.

**Adoptive transfer**. To characterization of cell division at early T$_{FH}$ differentiation, SMARTA CD4+ T cells were labeled with 5 µM of CTV (Invitrogen), and $2 \times 10^6$ of labeled Vα2+ SMARTA CD4+ cells were transferred followed by intravenously infected with $2 \times 10^6$ pfu of LCMV-Armstrong. To investigate the intrinsic effects with adoptive transfer model, $5 \times 10^6$ wild-type SMARTA cells were transferred into Ctrl and Mettl3$^{fl/fl}$Cd4-Cre host mice, followed by $2 \times 10^5$ pfu LCMV-Armstrong infection intraperitoneally.

**Bone marrow chimeric mice**. To generate bone marrow chimeric mice, lethally irradiated B6.SJL (CD45.1+) mice were transferred intravenously with a 1:1 mixture of $2.5 \times 10^6$ Mettl3$^{fl/fl}$Cd4-Cre (CD45.2+) and $2.5 \times 10^6$ B6.SJL (CD45.1+) bone marrow cells. After 8 weeks reconstitution, recipient mice were infected with LCMV-Armstrong strain.

**ELISA**. Analysis of the LCMV-specific antibody in serum was performed as previously described[53]. Briefly, lysates of LCMV-infected BHK-21 cells were used as substrate and LCMV-specific antibody (IgG) was titrated in serial dilutions of serum using HPR-conjugated goat-anti mouse IgG antibodies (Cat. no. A90-131P-39; 1:5,000; Bethyl laboratories).

**Immunofluorescence staining**. Tissue specimens submerged in OCT compound were quickly frozen in liquid nitrogen and cut into 10 µm thickness. Frozen tissue sections were then fixed in cold acetone for 30 min at −20 °C, blocked with 1% BSA and Fc-blocker (2.4G2; 1:100; BD Biosciences) in PBS. The tissue sections were then stained with biotinylated PNA (Cat. no. BA-0074; 1:20; Vector laboratories), followed by staining with BV510-labeled anti-CD4 (RM4-5; 1:100; BD Biosciences), APC-labeled anti-IgD (11-26c; 1:100; Thermo Fisher Scientific), and AF488-conjugated streptavidin (1:500; Invitrogen). Then the slides were washed at least three times with PBS. Coverslips were mounted on slides using an antifade kit (Beyotime Biotechnology) and then examined using an Andor Dragonfly confocal microscope. The images were processed with Imaris (v8.1; Bitplane) and Image J (v1.52 g; NIH).

**Retroviral vectors, transduction, and cell transfer**. Tcf7 (full-length CDS region of P45 isoform without 3′ UTR region[39]) and Mettl3 (wild-type and catalytic domain dead) coding sequences were amplified and cloned into the pMIG-R1 vector (MSCV-IRES-GFP). Retrovirus was packaged by transfection of HEK293T cells (Cat. no. CRL-3216; ATCC) with the retroviral vectors along with pCL$^{eco}$ plasmid. SMARTA cells were activated in vivo by injection of 200 µg of LCMV GP61-80 (GLNGPDIYKGVYQFKSVEFD) peptide into Mettl3$^{fl/fl}$Cd4-Cre SMARTA mice or their littermate wild-type transgenic mice. After 16~18 h, activated SMARTA cells were isolated, purified, and 'spin-infected' for 120 min at 37 °C by centrifugation ($1000 \times g$) with freshly harvested retroviral supernatants supplemented with 8 µg/mL of polybrene (Sigma-Aldrich), then cultured overnight in the presence of 20 ng/mL of IL-2 (Peprotech), and 250 nM of LCMV GP61-80. The spinofection was repeated the next day, and a total of 0.5~1 × 10^6 retroviral infected SMARTA CD4+ T cells were then adoptively transferred into recipient

mice, followed by infection of the hosts with $2 \times 10^5$ pfu LCMV-Armstrong within 24 h.

**RNA-seq and data analysis**. For isolation of T$_H$1 cells and T$_{FH}$ cells, splenocytes from Mettl3$^{fl/fl}$Cd4-Cre mice and their control littermates on day 8 post viral infection were subjected to depletion of cells positive for lineages markers by using biotin-conjugated antibodies (anti-B220 (RA3-6B2; 1:100), anti-CD8 (53-6.7; 1:100), anti-Gr.1 (RB6-8C5; 1:100), anti-CD11b (M1/70; 1:100), anti-CD11c (N418; 1:100), anti-TER119 (TER-119; 1:100), and anti-CD49b (DX5; 1:100); all from Thermo Fisher Scientific) coupled to Dynabeads M-280 Streptavidin (Invitrogen), followed by surface stained. CD44+SLAM$^{hi}$ T$_H$1 cells and CD44+SLAM$^{lo}$ T$_{FH}$ cells were sorted with a FACSAria II cell sorter (BD Biosciences) with FACSDiva software (v7.0) and subsequently lysed with TRIzol Reagent (Life technologies). Total RNAs were extracted and then subjected to Annoroad (Beijing, China) for library construction and RNA sequencing. The qualities of clean reads were assessed by FastQC (v0.11.5). Then the reads were mapped to mouse genome mm10 (version M17) using TopHat (v2.1.1). The read counts of all genes were estimated by HTseq (v0.6.1) and differentially expressed genes were identified by DESeq2 (v1.18.1). TPM, FPKM, and RPKM were calculated, and upregulated or downregulated genes in Mettl3$^{fl/fl}$Cd4-Cre T$_{FH}$ or T$_H$1 cells were identified by expression changes ≥ 2-fold and FDR < 0.01.

**Quantitative RT-PCR**. Total RNAs were extracted from sorted cells using RNeasy Mini Kit (Qiagen) followed by cDNA synthesis with FastQuant RT Kit (Tiangen). Quantitative RT-PCR was carried out with SuperReal PreMix Plus SYBR Green (Tiangen) on a CFX96 Connect™ Real-Time System (Bio-Rad). Results were processed by Microsoft Excel and then were normalized to the expression of Hprt1 transcripts. Fold differences in expression levels were calculated according to the $2^{-\Delta\Delta CT}$ method. All primers used are listed in Supplementary Table 1.

**Gene set enrichment analysis**. GSEA was performed with GSEA desktop software (v4.1.0) from the Broad Institute. The T$_{FH}$ gene set[11], GC T$_{FH}$ gene set[11], T$_H$1 gene set[12], T$_H$2 gene set[34], T$_H$17 gene set[35], and Treg gene set[34] have been described before. The 'TCF1-activated genes in T$_{FH}$ cells' gene set contains 569 genes that are downregulated by ≥1.5-fold in Tcf7$^{fl/fl}$Cd4-Cre T$_{FH}$ cells; The 'TCF1-suppressed genes in T$_{FH}$ cells' gene set contains 513 genes that are upregulated by ≥1.5-fold in Tcf7$^{fl/fl}$Cd4-Cre T$_{FH}$ cells (GSE65693).

**Luciferase reporter assay**. Tcf7 3′ UTR segment (~200 nt) was amplified from a mouse double-positive (DP) thymocytes cDNA library by PCR and inserted into the pGL4.23 vector (Promega) by using XbaI and FseI restriction sites. The Tcf7 3′ UTR mutation plasmids were generations from site-directed mutagenesis. All plasmids and mutations were verified by sequencing. HEK293T cells were seeded into 24-well plates in triplicate to allow 80% confluency in the next day. A total of 200 ng of reporter plasmids (Fluc) and 20 ng of Renilla luciferase (Rluc) control plasmids (pRL-TK) were co-transfected using Lipofectamine 2000 reagent (Invitrogen) under METTL3 overexpressing. Fluc and Rluc activities were measured 24 h later with the Dual-Luciferase Reporter Assay System (Promega) according to the manufacturer's instructions. The relative luciferase activity was calculated by dividing Fluc by Rluc and normalized to pGL4.23 empty vector for each assay.

**RNA decay assay**. CD4+ T cells were purified from Mettl3$^{fl/fl}$ER$^{T2}$-Cre mice and their control wild-type mice as described above. $5 \times 10^5$ purified CD4+ T cells in RPMI 1640 medium supplemented with 10% fetal bovine serum, 20 ng/mL of IL-2, 10 ng/mL of IL-7, and 5 µM of 4-Hydroxytamoxifen (Sigma-Aldrich) were seeded into 48-well plates. After 48 h, actinomycin D (MedChemExpress) was added to a final concentration of 5 µM, and cells were harvested at $t = 0, 1, 2$ h after actinomycin D treatment. Total RNAs were extracted and subjected to RT-qPCR analysis. Results were processed by Microsoft Excel and then normalized to the expression of Gapdh transcript. Fold differences in expression levels were calculated according to the $2^{-\Delta\Delta CT}$ method. All primers used are listed in Supplementary Table 1.

**m$^6$A-miCLIP-SMARTer-seq and data processing**. Total RNAs from CD4+ SMARTA cells sorted from LCMV GP61-80-primed Mettl3$^{fl/fl}$Cd4-Cre SMARTA or Ctrl SMARTA mice were extracted with TRIzol Reagent (Life Technologies). mRNAs were further isolated from total RNAs using Dynabeads mRNA purification kit (Ambion). The procedures of m$^6$A-miCLIP-SMARTer-seq were according to the previously reported methods with some modifications[54]. Briefly, 100 ng of mRNAs were fragmented to ~100 nt by using the fragmentation reagent (Life Technologies) and incubated with 5 µg of specific antibody against m$^6$A (Abcam) in 500 µL of immunoprecipitation buffer (50 mM Tris-HCl (pH 7.4), 100 mM NaCl, 0.05% NP-40) with gentle rotation at 4 °C for 2 h. The mixture was then transferred into a clear flat-bottom 96-well plate (Corning) on ice and irradiated three times with 0.15 J/cm$^{-2}$ at 254 nm in a CL-1000 Ultraviolet Crosslinker (UVP). The irradiated mixture was then transferred to a new tube and incubated with 50 µL of pre-washed Dynabeads Protein A (Life Technologies) at 4 °C for 2 h. After extensive washing twice with high-salt wash buffer (50 mM Tris-HCl

(pH 7.4), 1 M NaCl, 1 mM EDTA, 1% NP-40, 0.1% SDS) and twice with immunoprecipitation buffer, the mixture on beads was subjected to dephosphorylation with T4 PNK (NEB) for 20 min at 37 °C. After extensive washing, the RNA was eluted from the beads by proteinase K (Sigma-Aldrich) digestion at 55 °C for 1 h, followed by phenol-chloroform extraction and ethanol precipitation. The purified RNA was subjected to library construction using a SMARTer smRNA-Seq Kit for Illumina (Clontech Laboratories) according to the manufacturer's instructions. Sequencing was carried out on an Illumina X-ten platform. m6A-miCLIP-SMARTer-seq data (paired-end) were analyzed as previously described[55]. Adapter sequences at the 3′ ends were removed first. R2 reads were then transformed to reverse complementary sequences and merged with R1 reads. PolyA tails from the library were trimmed by Cutadapt (v1.17) and fastq2collapse (v1.1.3) was used to remove duplicated reads; afterward, the barcodes of reads were removed. Low-quality bases were discarded, and only reads longer than 18 nt were retained. The remaining reads were mapped to the reference genome (mm10) using BWA (v0.7.17) with the parameter: -n 0.06. The mutation information was extracted and PCR duplicates according to BWA's results were removed separately by parseAlignment.pl (−map-qual 1 −min-len 18) and tag2collapse.pl (-EM 30 −seq-error-model alignment). CIMS.pl was used to calculate the coverage ($n$) of the mutation site and transition number ($m$). Mutation sites with parameters, $m \geq 3$, $k/m \geq 0.01$, and $k/m \leq 0.5$, were kept. The remaining sites within the RRACH motif sequence were considered as m6A.

**m6A-RIP-qPCR**. Purified mRNAs of CD4+ SMARTA cells sorted from *Mettl3*fl/fl *Cd4*-Cre SMARTA or Ctrl SMARTA mice were prepared and fragmented into ~100 nt by RNA fragmentation reagents (Life Technologies). Immunoprecipitation was performed using anti-m6A antibody (Abcam) as described above. The enrichment of m6A was measured with quantitative RT-PCR. Primers for m6A-RIP-qPCR are listed in Supplementary Table 1.

**RIP-qPCR**. $5 \times 10^6$ CD4+ SMARTA T cells were isolated and lysed with 1 mL cell lysis buffer (150 mM KCl, 10 mM HEPES (pH 7.6), 2 mM EDTA, 0.5% NP-40, 0.5 mM DTT, 1:100 proteinase inhibitor cocktail, and 0.4 U/μL RNasin) at 4 °C for 30 min. After centrifugation, the supernatant (10% of which was kept as input) was subjected to RNA immunoprecipitation with anti-METTL3 (Abcam) coupled with Dynabeads Protein A (Life Technologies). RNA was isolated from the beads and input samples for RT-qPCR. Primers for RIP-qPCR are listed in Supplementary Table 1.

**Statistical analysis**. Statistical analysis was performed with Prism 8.0 (GraphPad). An unpaired two-tailed Student's *t* test with a 95% confidence interval, one-way ANOVA, or two-way ANOVA analysis was used to calculate *P* values.

**Reporting summary**. Further information on research design is available in the Nature Research Reporting Summary linked to this article.

## Data availability

RNA-seq and m6A-miCLIP-SMARTer-seq datasets have been deposited in Gene Expression Omnibus (GEO) under the accession number GSE129650. All data are available from the corresponding author upon reasonable request. Source data are provided with this paper.

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

## Acknowledgements

We thank Drs. Qi Zhou and Wei Li (Institute of Zoology, Chinese Academy of Sciences) for *Mettl3*fl/fl mice; Dr. Rafi Ahmed (Emory University) for SMARTA mice; Dr. Hai-Hui Xue (Hackensack University Medical Center) for constructive advice on the manuscript; Dr. Xinyuan Zhou (Third Military Medical University) for retroviral vectors; and members of Drs. Ye, Yang and Yu Laboratories for their technical advice and help. This work was supported in part by grants National Key Research and Development Program of China (2017YFA0104401 to S.Y.), National Natural Scientific Foundation of China (31970831, 31630038, 31571522, & 31422037 to S.Y.; 31825011 to L.Y.; 31430022 to Y.-G.Y.), the Project for Extramural Scientists supported by State Key Laboratory of Agrobiotechnology of China Agricultural University (2018SKLAB6-30, 2019SKLAB6-6, & 2019SKLAB6-7 to S.Y.), and the Youth Innovation Promotion Association (CAS 2018133 to Y.-G.Y.).

## Author contributions

Y. Yao and W.G. performed the overall experiments with Y. Yang and L.X. M.Y., Z.S., X.C., G.Y., Z.Q., Jingjing L., and T.Z. helped to carry out experiments and interpreted data. Juanjuan L. helped to breed B6.SJL mice. Y. Yang executed m6A-miCLIP-SMARTer-seq experiments. M.Y., Y.Z., and F.W. analyzed the RNA-seq and m6A-miCLIP-SMARTer-seq data. Y. Yao, W.G., and S.Y. were responsible for analysis of overall data. Y. Yao and S.Y. wrote the manuscript with the revision by all authors. S.Y., L.Y., and Y.-G.Y. conceived experiments and supervised the study.

## Competing interests

The authors declare no competing interests.
