## [Peer Review File · Nature Communications]

REVIEWER COMMENTS

Reviewer #1 (Remarks to the Author):

In this manuscript Yao et al analyze how mRNA N6-methyladenosine modification influences the development of follicular helper T (Tfh) cells. Although other mechanisms of post-transcriptional regulation like miRNAs have been analyzed before, this specific and important question is currently not known for Tfh cells. They show that METTL3-deficient mice, a methyltransferase central for mRNA methylation, almost completely lack Tfh cells in an LCMV infection model. They further demonstrate that this effect is T cell intrinsic and that ectopic overexpression of METTL3 in antigen-specific T cells can fully reverse the observed effects. A global analysis of methylated mRNAs identifies the transcription factor TCF-1 as a novel target of post-transcriptional regulation by methylation.

Although these mentioned data are all highly convincing, the central question, whether METTL3 really specifically regulates Tfh cell differentiation or whether METTL3 deficiency results in a general defect in T cell activation and differentiation as has been described before (Li et al., *Nature* 548:338 (2017)), remains unanswered. The authors do not analyze any other T cell subsets like Th2 or Th17 in their model. Moreover, the identified target TCF-1, although important for Tfh cells, does not specifically regulate this T cell subset but is also involved in differentiation of Th2, Th17, Treg and CD8+ effector T cells.

1) Although the authors designate the SLAMHi CXCR5^{low} cells in Fig. 1a as "TH1 cells", no experimental data are shown for this. To exclude that these cells are simply undifferentiated or even naive cells (depending on the gating, see comment 6), the authors have to stain for T-bet. Given the described role of TCF-1 for Th2, Th17 and Treg differentiation it would be also important to analyze these T cell subsets (which might not be possible in the LCMV infection model but in the KLH model used in Suppl. Fig. 2).

2) For the transcriptome analysis of METTL3-deficient versus wildtype littermates, the authors state that "sorted TFH cells" have been used. Any details, how cells were sorted and purity controls are missing. However, the central question is, why did the authors sort the cells? As shown in Fig. 1, METTL3-deficient mice essentially have no CXCR5+PD-1+ Tfh cells. Sorting of the few CXCR5+/PD-1+ positive cells or the SLAM^{lo} but CXCR5-negative population (see comment 7) most likely results in a highly impure population mainly consisting of non-Tfh cells. Thereby, it cannot be ruled out that the enrichment for a Tfh core signature shown in Fig. 4 is simply an artefact of the sorting procedure. The authors should specify their sorting strategy and show purity controls of sorted populations. However, the more convincing experiment would be to repeat the transcriptome analysis on unsorted cells. The authors could use their Smarta adoptive transfer model to sort for total antigen-specific cells. This experiment would reveal whether METTL3-deficient mice specifically lack a Tfh signature or also signatures for Th2, Th17 and Treg or simply generally lack all differentiated T cell subsets.

3) The authors claim that in their model METTL3-deficiency does not impact general T cell activation. The only data shown for this claim is Suppl. Fig. 4. However, these data are difficult to interpret. In fact, there is a difference in CD44 expression. CD69 seems to be negative in both experimental groups (day 3 might be too late for analysis). Isotype controls would help to determine which cells are positive for the marker.

4) METTL3-deficiency has a dramatic impact on the expansion of antigen-specific T cells (best seen in Fig. 5c). To conclude that the defects in B cell differentiation shown in Fig. 1 e-g are the result of impaired Tfh cell differentiation, the authors have to somehow correct for this overall lack of antigen-specific T cells (e.g. by transferring more Smarta cells in the METTL3-deficient group).

5) In the bone marrow chimera experiments (Fig. 2), the authors compare the *Mettl3* fl/fl Cd4-Cre population with the *Mettl3* wt/wt Cd4-Cre population in the control animals but not with the co-transferred WT competitor population. This is unusual and these data should be shown in addition.

6) In all figures, it remains unclear which cell populations are displayed. In the polyclonal system,

did the authors pre-gate for CD4⁺CD44^{high} cells? For the Smarta adoptive transfer system, the authors state, that "cells from recipient mice" are shown, however, this should be gated on antigen-specific cells from the donor mice present in the recipients. This should be described unequivocally.

7) In Fig. 1a, 2b, 5d the "SLAMF6⁺ CXCR5⁺ TFH cells" gate for the METTL3-deficient mice also includes many cells which are essentially CXCR5 negative. Staining for CXCR5/PD-1 and Bcl-6 indicates that these are not Tfh cells. The gate should be adjusted.

Minor points

8) Fig. 4c and d: NOM p-val and FDR cannot be 0.0. They are either < 0.01 or < 0.001 depending on the number of permutations.

9) Why does the Methods section contain a paragraph on immunohistology but no data are shown?

10) Suppl. Fig. 4 d,e (named e,f in legend): Is this really gated on Tfh cells (which markers?) or all CD4⁺ cells as suggested by the plots?

11) Line 368: this is not an anti-PNA antibody but in fact peanut agglutinin which binds to specific glycostructures on GC B cells.

Reviewer #2 (Remarks to the Author):

This is a well-done study that highlights the role of the Mettl3 methyltransferase and m6A RNA modifications in regulation of T cell differentiation, specifically that of Tfh cells. The authors find that T cell deletion of Mettl3 prevents Tfh differentiation in response to viral (LCMV) infection and immunization, that this is associated with decreased mRNA levels for Tcf7, Bcl6, CXCR5 and Icos and other genes important for Tfh cells. They further find that Tcf7 mRNA is modified by m6A (as is Bcl6), that mutation of the site in the 3' UTR of Tcf7 prevents m6A modification and that re-expression of Tcf7 can rescue Tfh generation. The data is solid, and the experiments seem for the most part well performed and described. However, one concern is how this reconciles with a recent paper from YC Liu evaluating the effects VHL-deficiency showing a major defect in Tfh differentiation that they attribute to increased m6A methylation of Icos mRNA. In that paper, the authors used shRNA to disrupt Mettl3 expression (although levels of Mettl3 were not shown), showing an increase in Tfh cell generation in response to LCMV. While there are many reasons why a knockdown may differ in phenotype from a knockout, the authors need to discuss and address these issues. These issues do not necessarily distract from their results, but should be discussed.

Specific comments:

TCF1 is expressed in naïve T cells, but in CD4 cells its expression is only maintained in Tfh cells in response to LCMV. Are Tcf7/TCF1 levels reduced in naïve cells or only specifically in Tfh cells from Mettl3^{CD4Cre} mice? Is TCF1 downregulated normally upon stimulation of T cells with anti-CD3/antigen plus IL-2 in culture as has been reported by the Reiner lab? It would be good to know whether this is specific for Tfh cells.

Was Icos modified by m6A? A recent paper from Yun Cai Liu's lab (Zhu et al JEM 2019) implicated increased m6A modification of Icos mRNA and decreased Icos expression, as well as increased Mettl3 expression as a reason why VHL-deficient CD4 cells fail to develop into Tfh cells. They further show that knockdown of Mettl3 increased Tfh differentiation in response to LCMV. These results suggest that m6A modification of mRNAs may affect multiple aspects of Tfh differentiation. While it is very likely that VHL affects multiple aspects of Tfh generation, these issues should be addressed. Did the authors look at m6A of Icos mRNA? The authors likely have these data and given this previous paper, the authors should examine this and discuss. This does not distract from their data, but is important to discuss/examine, given the previous paper.

The authors show rescue of the *Mettl3* phenotype by re-expression of *Mettl3*. Given the above cited paper from the Liu lab suggests that increased *Mettl3* expression impairs Tfh cell generation, the authors should show the level of *Mettl3* expression in the retroviral transduced cells, if possible.

Minor points:

The authors show rescue of Tfh cell generation with ectopic expression of *Tcf7*. I presume their retroviral construct did not include the 3' untranslated region, but this should be stated. Otherwise, it is not clear why they did not use the mutant version of *Tcf7* mRNA.

Supplemental Fig 3C: Although the schematic is helpful, please label blue and red ("Host") for clarity (I think that is correct). The results and description are a bit confusing. Why are the donor SMARTA (presumably all WT) Tfh and Th1 cell numbers decreased in the mutant host (even if not significant)?

In Figure 4D, please include a scale for the heatmap.

In summary, this is a nice paper, but needs to address or at least discuss some previous data in the literature.

Reviewer #3 (Remarks to the Author):

Yao and colleagues investigated the role of METTL3 and m6A in generation of T follicular helper cells (TFH) from CD4 cells by using mice with CD4-specific METTL3 deletion. They nicely showed that deletion of METTL3 in CD4 cells inhibited TFH cell differentiation and germinal center response through an intrinsic mechanism. METTL3 regulates the expression of TFH specific genes such as *Tcf7*, *Bcl6*, *Icos*, and *Cxcr5* through its methyltransferase activity. m6A modification of *Tcf7* leads to increased stability of *Tcf7* transcripts. This manuscript provided a new functional role of m6A and METTL3 in regulating TFH cell differentiation and implications in humoral immunity. However, the following comments need to be addressed.

Major comments:

1. The authors discussed the possible m6A readers for m6A-modified *Tcf7* and indicated that the possible readers such as IGF2BPs are expressed at low levels in TFH cells. However, protein levels of IGF2BPs were not looked at. In addition, since TFH cells are generated from CD4 cells, therefore, the levels and roles of IGF2BPs in regulating *Tcf7* expression in CD4 cells need to be assessed.
2. Figure 6f and 6g: upon METTL3 deletion, the m6A enrichment on *Tcf7* transcripts were decreased. However, the total RNA levels of *Tcf7* were also decreased in METTL3 deleted cells. Normalized peak distribution needs to be shown to demonstrate the effect of METTL3 deletion on specific m6A peaks across the *Tcf7* transcripts.
3. Figure 6j: It seemed that METTL3-Myc increased the luciferase activity of EV. Does this due to transfection efficiency difference or other effect of METTL3 on EV activity? Does this also contribute to the effect of METTL3 on wild-type reporter. This needs to be addressed.
4. Figure 6k and 6l: the effect of METTL3 deletion on the mRNA stability of *Tcf7* is moderate. Can other mechanism be involved in the regulation of *Tcf7* stability and expression by METTL3?
5. Does METTL3 interact with the *Tcf7* transcripts?
6. The effect of METTL3 deletion on m6A enrichment in the *Tcf7* gene needs to be validated by m6A IP qPCR analysis.
7. The importance of TFH cell differentiation regulated by METTL3 can be further assessed. For example, does adoptive transfer of TFH cells from WT mice rescue the defect in CD4-specific METTL3 deletion mice in response to viral infection?

Minor comments:

1. Line 264: typo in "MRTTL3"
2. Line 302-303: sentence needs correction. "Does is....".

Point-by-point responses to reviewers

Reviewer #1 (Remarks to the Author):

In this manuscript Yao et al analyze how mRNA N6-methyladenosine modification influences the development of follicular helper T (Tfh) cells. Although other mechanisms of post-transcriptional regulation like miRNAs have been analyzed before, this specific and important question is currently not known for Tfh cells. They show that METTL3-deficient mice, a methyltransferase central for mRNA methylation, almost completely lack Tfh cells in an LCMV infection model. They further demonstrate that this effect is T cell intrinsic and that ectopic overexpression of METTL3 in antigen-specific T cells can fully reverse the observed effects. A global analysis of methylated mRNAs identifies the transcription factor TCF-1 as a novel target of post-transcriptional regulation by methylation.

Although these mentioned data are all highly convincing, the central question, whether METTL3 really specifically regulates Tfh cell differentiation or whether METTL3 deficiency results in a general defect in T cell activation and differentiation as has been described before (Li et al., Nature 548:338 (2017)), remains unanswered. The authors do not analyze any other T cell subsets like Th2 or Th17 in their model. Moreover, the identified target TCF-1, although important for Tfh cells, does not specifically regulate this T cell subset but is also involved in differentiation of Th2, Th17, Treg and CD8⁺ effector T cells.

We sincerely thank the reviewer for evaluating our manuscript and for praising the quality of our data. We do appreciate the reviewer for giving us constructive suggestions to improve the quality of our MS. Following the reviewers' insightful comments, we have performed extensive new experiments and extended the scope of our study during the past several months. We believe that the manuscript is greatly improved with more convincing and solid data than previous version. For specific concerns, we provided point to point responses as follow.

1) Although the authors designate the SLAMF6^{hi} CXCR5^{low} cells in Fig. 1a as "TH1 cells", no experimental data are shown for this. To exclude that these cells are simply undifferentiated or even naive cells (depending on the gating, see comment 6), the authors have to stain for T-bet. Given the described role of TCF-1 for Th2, Th17 and Treg differentiation it would be also important to analyze these T cell subsets (which might not be possible in the LCMV infection model but in the KLH model used in Suppl. Fig. 2).

We thank the reviewer for this suggestion. Upon acute viral infection, virus-specific naïve CD4⁺ T cells generally differentiate into T_H1 cells or T_{FH} cells (Hale et al., 2013; Marshall et al., 2011; Xu et al., 2015). We also analyzed T_H1 populations on day 8 post LCMV infection. In this study, we

mainly focused on the role of METTL3 in T_{FH} cells with solid experimental evidences by using LCMV and KLH immunization models. Our data indicated that ablation of METTL3 resulted in more severe defects in T_{FH} cells (21.4 fold change) than T_{H1} cells (1.9 fold change) upon acute viral infection (**Figure 1b**). Given TCF-1 as a major downstream effector is substantially downregulated in METTL3-deficient CD4⁺ T cell upon acute infection, the effects at distinct degree in T_{FH} cells compared with T_{H1} cells seem logically reasonable, which corresponds to the previous findings that TCF-1 is critical for T_{FH} cell differentiation, but not essential for T_{H1} cells.

To address the concern ‘these cells are even naïve cells’, we analyzed the CD44^{hi}CD62L^{lo}CD4⁺ cells on day 8, and found both WT and *Mettl3*^{fl/fl}*Cd4-Cre* CD4⁺ T cells activation was normal (47.2% ± 7.5% Ctrl versus 46.7% ± 4.6% *Mettl3*^{fl/fl}*Cd4-Cre*; Figure R1). As requested, we also analyzed T-bet expression on CD62L⁺CD44^{lo}CD4 cells (naïve), CD44⁺CXCR5⁺ T_{FH} cells, and CD44⁺CXCR5⁻ T_{H1} cells (**Supplementary Figure 1b**). We observed that CD44⁺CXCR5⁻ T_{H1} cells expressed much higher levels of T-bet than CD44⁺CXCR5⁺ T_{FH} and naïve cells, which is consistent with published data (Hale et al., 2013). Moreover, our results suggested that both T_{H1} cells and T_{FH} cells downregulated T-bet expression in the absence of METTL3 (**Supplementary Figure 1b**). The text was amended accordingly.

Figure R1. Analysis of CD44^{hi}CD62L^{lo}CD4⁺ cells

As requested, we also analyzed other T helper lineages differentiation using the KLH model. We found that both GATA3⁺ and IL-4-producing cells (mainly T_{H2} cells) as well as Foxp3⁺ cells (mainly Treg cells) were not altered due to the absence of METTL3 (**Supplementary Figure 2e-h**), indicating T_{H2} and Treg cell differentiation are not affected in *Mettl3*^{fl/fl}*Cd4-Cre* mice upon KLH immunization. Whereas, both RORγt⁺ and IL-17a-producing cells (mainly T_{H17} cells) were significantly decreased in METTL3-deficient mice (**Supplementary Figure 2e-h**), revealing METTL3 is essential for T_{H17} cell differentiation *in vivo* upon KLH immunization. The text was amended accordingly. However, we found that most of the conclusions describing roles of TCF-1 in T_{H2}, T_{H17}, and Treg differentiation were typically drew from *in vitro* stimulation and culture system as well as corresponding mice models after we learned the relative articles carefully (van Loosdregt et al., 2013; Xing et al., 2019; Yang et al., 2019; Yu et al., 2009; Zhang et al., 2018). Based on results from our and other groups (Li et al., 2017; Tong et al., 2018), we thought KLH immunization model *in vivo* is not an optimizing approach to investigate T_{H2}, T_{H17} and Treg differentiation. LCMV and KLH immunization models are widely applied to investigate Tfh cells.

2) For the transcriptome analysis of METTL3-deficient versus wildtype littermates, the authors state that "sorted TFH cells" have been used. Any details, how cells were sorted and purity controls are

missing. However, the central question is, why did the authors sort the cells? As shown in Fig. 1, METTL3-deficient mice essentially have no CXCR5+PD-1+ Tfh cells. Sorting of the few CXCR5+/PD-1+ positive cells or the SLAM^{lo} but CXCR5-negative population (see comment 7) most likely results in a highly impure population mainly consisting of non-Tfh cells. Thereby, it cannot be ruled out that the enrichment for a Tfh core signature shown in Fig. 4 is simply an artefact of the sorting procedure. The authors should specify their sorting strategy and show purity controls of sorted populations. However, the more convincing experiment would be to repeat the transcriptome analysis on unsorted cells. The authors could use their Smarta adoptive transfer model to sort for total antigen-specific cells. This experiment would reveal whether METTL3-deficient mice specifically lack a Tfh signature or also signatures for Th2, Th17 and Treg or simply generally lack all differentiated T cell subsets.

We thank the reviewer for raising this question. We are sorry for the ambiguous description of the sorting strategy. Considering activated CD4⁺ T cells generally differentiate into either T_{H1} or T_{FH} cells during LCMV infection (Hale et al., 2013; Marshall et al., 2011; Xu et al., 2015), and loss of METTL3 affects CXCR5 expression, we applied SLAM and CD44 combination to distinguish CD44⁺SLAM^{hi} T_{H1} cells and CD44⁺SLAM^{lo} T_{FH} cells (assume as potential or defective T_{FH} cells) and subjected to RNA-seq, respectively. These two subsets covered all CD44⁺CD62L^{lo} activated CD4⁺ T cells (in accordance with the reviewer's suggestion "transcriptome analysis on unsorted cells") and this strategy ensured the non-T_{H1} population unsupervised. The detailed sorting strategy and purity controls are shown in **Supplementary Figure 7**.

Owing to acute viral model employed, it is hard to see other T helper lineages in activated CD4⁺ T cells, except T_{H1} and T_{FH} cells (Hale et al., 2013; Marshall et al., 2011; Xu et al., 2015). Based on our sorting strategy (T_{H1} and T_{FH}-like subsets covered the whole activated CD4⁺ T cell populations after infection), we supplemented T_{H1} RNA-seq data and performed GSEA analysis with distinct T helper lineage signature genes together with T_{FH} cells. We observed several signature genes related to T_{H1}, T_{H2}, T_{H17}, and Treg were all enriched in METTL3-deficient T_{FH} cells. Similarly, these signature genes were also enriched in METTL3-deficient T_{H1} cells. These results suggested that loss of METTL3 leads to more or less disordered gene profiles of T_{FH} and T_{H1} transcription program (**Supplementary Figure 5b and c**). It is worth mentioning that Li et al. have reported that METTL3-deficient naïve CD4⁺ T cells differentiated into fewer T_{H1} and T_{H17} cells and more T_{H2} cells by using *in vitro* culture system (Li et al., 2017), which is a well-documented approach for studying T helper lineage differentiation. Based on their data and our results from KLH model, we thought METTL3 deficiency might also affect other T helper lineages differentiation.

3) The authors claim that in their model METTL3-deficiency does not impact general T cell activation. The only data shown for this claim is Suppl. Fig. 4. However, these data are difficult to interpret. In fact, there is a difference in CD44 expression. CD69 seems to be negative in both

experimental groups (day 3 might be too late for analysis). Isotype controls would help to determine which cells are positive for the marker.

We thank the reviewer for his/her suggestion. We acknowledged that our previous result exhibited a modest significance in CD44 expression. Hence, we carefully checked our all data from three independent experiments, and found there was no obvious difference in CD44 expression between two genotypes by pooling all data together ($n = 9$ per group). CD69 is also detectable on day 3 in comparison with isotype and showed comparable expression between Ctrl and METTL3-deficient groups (**Supplementary Figure 4b, c**). The similar data presentations were also shown by other investigators (Choi et al., 2011; Vinuesa et al., 2005).

To further validate the conclusion that METTL3 deficiency does not alter T cell activation, we also isolated naïve CD4⁺ T cells from *Mettl3^{fl/fl}Cd4-Cre* and Ctrl SMARTA mice. The cells were cultured *in vitro* with/without anti-CD3/CD28 stimulation. On day 1, day 2, and day 3, the expression levels of CD25, CD44, and CD69 were detected. Our results indicated that *Mettl3^{fl/fl}Cd4-Cre* CD4⁺ T cells expressed comparable level of CD25, CD44, and CD69 compared with Ctrl cells (**Supplementary Figure 4d**). Taken together, our data suggested the overall activation of CD4⁺ T cells is normal in the absence of METTL3 upon acute viral infection.

4) METTL3-deficiency has a dramatic impact on the expansion of antigen-specific T cells (best seen in Fig. 5c). To conclude that the defects in B cell differentiation shown in Fig. 1 e-g are the result of impaired Tfh cell differentiation, the authors have to somehow correct for this overall lack of antigen-specific T cells (e.g. by transferring more Smarta cells in the METTL3-deficient group).

We thank the reviewer for raising this constructive suggestion. To achieve this goal, we transfer 5 million SMARTA cells into Ctrl or *Mettl3^{fl/fl}Cd4-Cre* mice to evaluate the impact of impaired T_{FH} differentiation on humoral response as described before (Vaeth et al., 2016). On day 8 post infection, we found the cell numbers of both GC B cells and plasma cells in *Mettl3^{fl/fl}Cd4-Cre* mice were comparable with Ctrl mice. Correspondingly, the PNA⁺ GC within B cell follicles were detectable in the spleens of *Mettl3^{fl/fl}Cd4-Cre* host mice. Meanwhile, the level of antigen-specific IgG in *Mettl3^{fl/fl}Cd4-Cre* host mice (transferred with SMARTA cells) showed no obvious difference in comparison with Ctrl host mice, while *Mettl3^{fl/fl}Cd4-Cre* mice (no SMARTA cells transfer) still showed defects in the production of LCMV-specific IgG. Collectively, these results indicated that extra T_{FH} cells could rescue the defects in B cells of *Mettl3^{fl/fl}Cd4-Cre* mice, emphasizing the notion that the defects in B cell differentiation which caused by impaired T_{FH} differentiation. These results are shown in **Figure 2d-g**, and the text was modified accordingly.

5) In the bone marrow chimera experiments (Fig. 2), the authors compare the *Mettl3 fl/fl Cd4-Cre* population with the *Mettl3 wt/wt Cd4-Cre* population in the control animals but not with the co-

transferred WT competitor population. This is unusual and these data should be shown in addition.

We thank the reviewer for raising this point. As requested, we compared the METTL3-deficient group with co-transferred WT competitor population, and the results are shown in current **Supplementary Figure 2**. The results from *Mettl3^{fl/fl}Cd4-Cre* V.S. co-transferred WT competitor are consistent with those from *Mettl3^{fl/fl}Cd4-Cre* V.S. *Mettl3^{wt/wt}Cd4-Cre*, and the conclusion remains unchanged. We replaced the whole figure panel according to the reviewer's suggestion and modified the description in relative sections.

6) In all figures, it remains unclear which cell populations are displayed. In the polyclonal system, did the authors pre-gate for CD4⁺CD44^{high} cells? For the Smarta adoptive transfer system, the authors state, that "cells from recipient mice" are shown, however, this should be gated on antigen-specific cells from the donor mice present in the recipients. This should be described unequivocally.

We agreed with the reviewer for this concern. Indeed, in the polyclonal system, we pre-gated CD4⁺CD44⁺CD62L⁻ cells for further analysis. For adoptive transfer model, we analyzed T_{FH} cells gated on antigen-specific CD4⁺CD44⁺ SMARTA cells (all activated cells). To better address this concern, we show all gating strategy in **Supplementary Figure 7**. Meanwhile, we also modified corresponding legends with more detailed information for a better understanding of the gating strategies. We are sorry for the unclear statement "cells from recipient mice", and we modified the description as follow: "SMARTA CD4⁺ T cells from recipient mice".

7) In Fig. 1a, 2b, 5d the "SLAMf^{lo} CXCR5⁺ TFH cells" gate for the METTL3-deficient mice also includes many cells which are essentially CXCR5 negative. Staining for CXCR5/PD-1 and Bcl-6 indicates that these are no Tfh cells. The gate should be adjusted.

We agreed with the reviewer for his/her concern and thank the reviewer for this constructive suggestion. METTL3 deficiency affects CXCR5 expression, hence T_{FH} cells were first determined based on CXCR5 and SLAM combination in Ctrl cells, the same gate was directly applied to other experimental conditions in each set of experiments. We considered the METTL3-deficient cells fell in this gate is a population of defective T_{FH} cells from *Mettl3^{fl/fl}Cd4-Cre* mice. To avoid including potential CXCR5⁻ cells based on the gating strategies by using CXCR5 and SLAM combination, the gating strategy by using CXCR5 and CD44 combination were applied to distinguish T_{FH} cells (CD44⁺CXCR5⁺) and T_{H1} cells (CD44⁺CXCR5⁻), which is also widely used by other T_{FH} investigators (Barbet et al., 2018; Hao et al., 2018; Xu et al., 2015). The related contour plots of flow cytometry data and their statistic plots (Figure 1a, b; Figure 2b, c; Figure 5e, f; Figure 7c,d; Supplementary Figure 2a, b) were replaced accordingly.

Minor points

8) Fig. 4c and d: NOM p-val and FDR cannot be 0.0. They are either < 0.01 or < 0.001 depending on the number of permutations.

We thank the reviewer for this suggestion, and we modified it accordingly.

9) Why does the Methods section contain a paragraph on immunohistology but no data are shown?

We thank the reviewer for this concern. During the initial submission, we thought the immunohistology results are only support information for defective GC responses in *Mettl3^{fl/fl}Cd4-Cre* mice. Therefore, we did not present these results but left the methods section about Immunofluorescence staining. In current submission, we added these results back and showed them in **Figure 1g**. Moreover, we also added new immunohistology data from adoptive transfer model to show the rescue of defects in GC response (address major concern 4) in **Figure 2f**. We believe these pairs of correlative data are convincing to support the conclusion that the defects in B cell differentiation are caused by a dramatic impact on the expansion of T_{FH} cells due to METTL3 deficiency. We thank the reviewer for raising this point again.

10) Suppl. Fig. 4 d,e (named e,f in legend): Is this really gated on Tfh cells (which markers?) or all CD4+ cells as suggested by the plots?

We thank the reviewer for his/her concern. First, we are sorry for the labeling mistake of figure legend. Second, we analyzed the Caspase-3⁺ cells gated on CXCR5⁺CD25⁻ nascent T_{FH} cells (Li et al., 2018) but not just antigen-specific CD4⁺ T cells. In the former submission, we did not show the analysis of T_{FH} cells based on CXCR5 and CD25 expression. To avoid this unclear classification, we also showed this analysis (current **Supplementary Figure 4e, f**) and amended the legend for caspase-3 assay in current version.

11) Line 368: this is not an anti-PNA antibody but in fact peanut agglutinin which binds to specific glycostructures on GC B cells.

We are sorry for this mistake and modified the description accordingly.

Reviewer #2 (Remarks to the Author):

This is a well-done study that highlights the role of the Mettl3 methyltransferase and m⁶A RNA modifications in regulation of T cell differentiation, specifically that of Tfh cells. The authors find that T cell deletion of Mettl3 prevents Tfh differentiation in response to viral (LCMV) infection and immunization, that this is associated with decreased mRNA levels for Tcf7, Bcl6, CXCR5 and Icos and other genes important for Tfh cells. They further find that Tcf7 mRNA is modified by m⁶A (as is Bcl6), that mutation of the site in the 3' UTR of Tcf7 prevents m⁶A modification and that re-expression of Tcf7 can rescue Tfh generation. The data is solid, and the experiments seem for the most part well performed and described. However, one concern is how this reconciles with a recent paper from YC Liu evaluating the effects VHL-deficiency showing a major defect in Tfh differentiation that they attribute to increased m⁶A methylation of Icos mRNA. In that paper, the authors used shRNA to disrupt Mettl3 expression (although levels of Mettl3 were not shown), showing an increase in Tfh cell generation in response to LCMV. While there are many reasons why a knockdown may differ in phenotype from a knockout, the authors need to discuss and address these issues. These issues do not necessarily distract from their results, but should be discussed.

We thank the reviewer for considering our work as “a well-done study” and evaluating it as “The data is solid, and the experiments seem for the most part well performed and described”. Indeed, we noticed the recent paper from YC Liu that mentioned by reviewer has also referred to the effects of m⁶A methylation on T_{FH} cells. However, the relative data were generated by using an shRNA knockdown approach which might lead to the difference in phenotype of knockout mice model with many reasons as mentioned by reviewer, such as off-targets or non-specific effects at distinct regulatory layers. We agreed with the reviewer for his/her reasonable understanding of the results gained from different experimental systems and we discussed these issues in the relative section. We believe that our data generated by using multiple genetic mice and transfer models are solid and convincing, whereas “the differences” from two studies may reflect in the complex regulatory mechanism of m⁶A modification, alternatively, the different experimental approaches employed in the two studies may also contribute to the varies.

Specific comments:

1) TCF1 is expressed in naïve T cells, but in CD4 cells its expression is only maintained in Tfh cells in response to LCMV. Are Tcf7/TCF1 levels reduced in naïve cells or only specifically in Tfh cells from Mettl3 CD4Cre mice? Is TCF1 downregulated normally upon stimulation of T cells with anti-CD3/antigen plus IL-2 in culture as has been reported by the Reiner lab? It would be good to know whether this is specific for Tfh cells.

We agreed with the reviewer for this concern. The *Tcf7* mRNA (1.6-fold) and TCF-1 (2.3-fold) levels were also decreased in naïve CD4⁺ T cells from *Mettl3^{fl/fl}Cd4-Cre* mice except in T_{FH} cells (3.9-fold for *Tcf7* mRNA level; 4.7-fold for TCF-1 protein level) (Figure R2). TCF-1 is known as a crucial transcription factor for T_{FH} cells and its expression level is substantially elevated during T_{FH} cell differentiation in comparison with naïve CD4⁺ T cells (Xu et al., 2015). Given naïve CD4⁺ T cells from *Mettl3^{fl/fl}Cd4-Cre* mice express a lower level of TCF-1, the failure of TCF-1 upregulation in CD4⁺ T cells attributes to defects in T_{FH} cell differentiation. In our study, we found that METTL3-deficient T_{FH} cells express a much lower level of both *Tcf7* mRNA and TCF-1 protein upon LCMV infection, which has been approved as a major reason for defects in T_{FH} cells differentiation through the following series analyses.

Figure R2. Analysis of *Tcf7* mRNA and TCF-1 protein in naïve and T_{FH} cells

To address the concern “Is TCF1 downregulated normally upon stimulation of T cells with anti-CD3/antigen plus IL-2 in culture as has been reported by the Reiner lab?”, we isolated naïve CD4⁺ T cells from both *Mettl3^{fl/fl}Cd4-Cre* and Ctrl mice, and cultured *in vitro* for 3 days under anti-CD3/CD28 plus IL-2 condition. Our results indicated that both *Mettl3^{fl/fl}Cd4-Cre* and Ctrl CD4⁺ T cells downregulated *Tcf7* expression on day 3 (Figure R3), which is consistent with previous reports that CD4 T cells receiving IL-2 expressed less TCF-1 (Nish et al., 2017; Wu et al., 2015). Particularly, IL-2 is a major factor contributing to the downregulation expression of TCF-1 in this cultural system, because IL-2 treatment induced the expression of Blimp-1, and partially suppressed the expression of *Tcf7* (Wu et al., 2015). Collectively, we thought the downregulated TCF-1 upon stimulation of T cells with anti-CD3/CD28 plus IL-2 in culture is not strongly correlated with the role of TCF-1 in T_{FH} cells upon LCMV infection *in vivo*, indicating the specific mechanism associated with TCF-1 in our current study.

Figure R3. Analysis of *Tcf7* expression

2) Was *Icos* modified by m⁶A? A recent paper from Yun Cai Liu's lab (Zhu et al JEM 2019) implicated increased m⁶A modification of *Icos* mRNA and decreased *Icos* expression, as well as increased *Mettl3* expression as a reason why VHL-deficient CD4 cells fail to develop into Tfh cells. They further show that knockdown of *Mettl3* increased Tfh differentiation in response to LCMV. These results suggest that m⁶A modification of mRNAs may affect multiple aspects of Tfh differentiation. While it is very likely that VHL affects multiple aspects of Tfh generation, these issues should be addressed. Did the authors look at m⁶A of *Icos* mRNA? The authors likely have these data and given this previous paper, the authors should examine this and discuss. This does not distract from their data, but is important to discuss/examine, given the previous paper.

We thank the reviewer for raising this point. Zhu et al. reported that induced GAPDH protein by VHL deficiency reduced *Icos* expression through METTL3/METTL14-catalyzed m⁶A modification on *Icos* mRNA, indicating that elevated m⁶A modification on *Icos* mRNA in VHL deficient cells reduces *Icos* expression which is associated with attenuated T_{FH} cell differentiation (Zhu et al., 2019). In current study, we also observed m⁶A modification on *Icos* mRNA (Supplementary Figure 6a; Figure R4A). However, we found the m⁶A level on *Icos* mRNA was significantly decreased in *Mettl3^{fl/fl}Cd4-Cre* cells by m⁶A-RIP-qPCR (Figure R4B). Due to METTL3-deficient cells had lower expression of *Icos*, indicating m⁶A may positively regulate *Icos* expression, which is different with the previous report (Zhu et al., 2019). Collectively, m⁶A modification on mRNAs may affect multiple aspects of T_{FH} differentiation, and the underline mechanism illustrating the differences needs to be further investigated in the future. Relative results regarding ICOS and differences between these two studies have been discussed in Discussion section.

Figure R4. Characterization of m⁶A expression on *Icos* mRNA

3) The authors show rescue of the *Mettl3* phenotype by re-expression of *Mettl3*. Given the above cited paper from the Liu lab suggests that increased *Mettl3* expression impairs Tfh cell generation, the authors should show the level of *Mettl3* expression in the retroviral transduced cells, if possible.

We agreed with the reviewer for this concern. Actually, the METTL3 expression in GFP⁺CD4⁺ donor cells from recipients has been examined and the results were originally shown in Supplementary Figure 5a in initial submission, which indicated that the METTL3 expression in *Mettl3^{fl/fl}*Cd4-Cre mice was rectified upon the introduction of METTL3. In revised manuscript, we reorganized the entire data set and presented the relative data in current **Figure 5c** for readers' convenience. The results are also shown as follow (Figure R5).

Figure R5. Analysis of METTL3 expression

Minor points:

1) The authors show rescue of Tfh cell generation with ectopic expression of *Tcf7*. I presume their retroviral construct did not include the 3' untranslated region, but this should be stated. Otherwise, it is not clear why they did not use the mutant version of *Tcf7* mRNA.

We thank the reviewer for raising this question. As presumed by reviewer, the retroviral construct of *Tcf7* was a 1260 bp full length CDS without 3' untranslated region and we modified the relative description for better comprehension in revised version. Which kinds of *Tcf7* should be used for rescue experiments is a really good point. Initially, we thought about any possible design for the retroviral constructs before we carried out rescue experiments. However, we chose the full length CDS of *Tcf7* for ensuring simple and direct design of the experimental system. Two possible reasons existed in this investigation. On one side, compared with the ectopic expression of full length CDS of *Tcf7*, the constructs with WT/Mut 3' untranslated region might cause little effects by using retrogenetic approach, leading to no significant difference on phenotypes of T_{FH} cells by using three distinct constructs. On the other side, due to the m⁶A modification site exists in the 3' untranslated region, but not in any exon, it was really hard to reflect the difference caused by distinct 3' untranslated region with WT *Tcf7* or mutant construct (might have equal effect with full length CDS without any 3' untranslated regulatory elements). Based on reasons above, we simply performed the rescue experiments with full length CDS of TCF-1, and directly rescued the phenotype of METTL3-deficient T_{FH} cells by restoring the expression level of reduced TCF-1. These data supported the conclusion that METTL3 sustains T_{FH} differentiation via regulating *Tcf7* expression.

2) Supplemental Fig 3C: Although the schematic is helpful, please label blue and red (“Host”) for clarity (I think that is correct). The results and description are a bit confusing. Why are the donor SMARTA (presumably all WT) Tfh and Th1 cell numbers decreased in the mutant host (even if not significant)?

We thank the reviewer for his/her suggestion. To further illustrate the cell-intrinsic role of METTL3 in T_{FH} cell differentiation, we performed relative assay with an adoptive transfer model. WT CD45.1⁺ SMARTA cells (donors) were transferred into both CD45.2⁺ Ctrl and *Mettl3*^{fl/fl}*Cd4*-Cre Host mice and analyzed on indicated time points post LCMV infection. Our results indicated that the WT SMARTA exhibited similar T_{FH} and T_H1 cell differentiation, albeit the cell numbers of both cell types showed modestly but not statistically significant change (we presumed decreased total splenocytes of METTL3-deficient mice and individual differences contributed to the relative lower cell numbers, but not significant). As requested by Reviewer 1 (major concern 7), we changed our gating strategy by using CXCR5 and CD44 combination and modified the description in relative sections (**Figure 2**), but the conclusion remains consistent. To clarify the information more clearly, we also added ‘Host’ for both legends as ‘Ctrl Host’, ‘*Mettl3*^{fl/fl}*Cd4*-Cre Host’ and modified the relative description in the MS for this experimental design. We thank the reviewer again for this kind reminder.

3) In Figure 4D, please include a scale for the heatmap.

We are sorry for this missing and we added a scale for the heatmaps in current version accordingly.

4) In summary, this is a nice paper, but needs to address or at least discuss some previous data in the literature.

We thank the reviewer again for his/her praise on our MS. As requested, we provided new data sets illustrating m⁶A modification on *Icos* mRNA, and discussed previous data on ICOS reported by Liu Lab. The relative part in discussion was supplemented accordingly.

Reviewer #3 (Remarks to the Author):

Yao and colleagues investigated the role of METTL3 and m6A in generation of T follicular helper cells (TFH) from CD4 cells by using mice with CD4-specific METTL3 deletion. They nicely showed that deletion of METTL3 in CD4 cells inhibited TFH cell differentiation and germinal center response through an intrinsic mechanism. METTL3 regulates the expression of TFH specific genes such as Tcf7, Bcl6, Icos, and Cxcr5 through its methyltransferase activity. M6A modification of Tcf7 leads to increased stability of Tcf7 transcripts. This manuscript provided a new functional role of m6A and METTL3 in regulating TFH cell differentiation and implications in humoral immunity. However, the following comments need to be addressed.

We sincerely thank the reviewer for taking time to evaluate our manuscript, for appreciating the importance of this study, and for the valuable feedback provided, which has helped us improve this manuscript. We have addressed all concerns from reviewers with new experimental data, bioinformatics analysis, and in-depth discussion. Therefore, we believe the manuscript is more than improved as a result.

Major comments:

1. The authors discussed the possible m6A readers for m6A-modified Tcf7 and indicated that the possible readers such as IGF2BPs are expressed at low levels in TFH cells. However, protein levels of IGF2BPs were not looked at. In addition, since TFH cells are generated from CD4 cells, therefore, the levels and roles of IGF2BPs in regulating Tcf7 expression in CD4 cells need to be assessed.

We thank the reviewer for this constructive advice. To illustrate the expression level of IGF2BPs, we first retrieved the FPKM values from our RNA-seq and published RNA-seq data (Choi et al., 2015), and found the expression levels of *Igf2bp1*, *Igf2bp2*, and *Igf2bp3* were extremely low in T_{FH} cells (Table R1; *Ythdf2* transcript, which mediates degradation of m⁶A-methylated mRNA, was used as control), and the mRNA expression of these IGF2BPs was also validated by qPCR (Figure R6A). To further address this concern, we also performed Western Blotting in CD4⁺ T cells together with HEK293T cells as a positive control with antibodies against IGF2BPs as previously described (Huang et al., 2018). Our results indicated that protein levels of IGF2BPs in CD4⁺ T cells were too low to be detected (**Supplementary Figure 8**; Figure R6B). Based on these results, we considered that it is difficult to establish connections between IGF2BPs and the regulation of TCF-1 expression level in the current stage. Therefore, we cannot conclude who is the possible reader protein involved in the regulation of TCF-1 via m⁶A modification at the current stage.

Table R1. FPKM value of IGF2BPs

Source	Gene Name	Sample 1	Sample 2	Sample 3
This paper	Igf2bp1	0.05	0	0
This paper	Igf2bp2	0.14	0	0.02
This paper	Igf2bp3	7.82	11.18	13.33
This paper	Ythdf2	46.73	54.61	56.12
Choi et al., 2015	Igf2bp1	1.78	0.27	/
Choi et al., 2015	Igf2bp2	2.45	0.65	/
Choi et al., 2015	Igf2bp3	2.09	2.73	/
Choi et al., 2015	Ythdf2	20.27	19.39	/

Figure R6. Analysis of expression of IGF2BPs by qPCR and Western Blotting

2. Figure 6f and 6g: upon METTL3 deletion, the m6A enrichment on *Tcf7* transcripts were decreased. However, the total RNA levels of *Tcf7* were also decreased in METTL3 deleted cells. Normalized peak distribution needs to be shown to demonstrate the effect of METTL3 deletion on specific m6A peaks across the *Tcf7* transcripts.

We thank the reviewer for providing us this constructive suggestion. In the initial submission, the count numbers from both RNA-seq and miCLIP-seq were used for miCLIP-seq analysis to eliminate the effects of expression level, but the IGV plots did not exhibit normalized peak distribution. As requested, we re-analyzed our m⁶A-miCLIP-seq data and used bamCompare of deepTools package to normalize the peak distribution as described before (Ramirez et al., 2016), a tool can be used to generate a bigWig or bedGraph file based on two BAM files that are compared to each other while being simultaneously normalized for sequencing depth. The IGV plots exhibiting *Tcf7* and other mRNAs by using new analysis are shown in **Figure 6f** and **Supplementary Figure 6a**, respectively.

3. Figure 6j: It seemed that METTL3-Myc increased the luciferase activity of EV. Does this due to transfection efficiency difference or other effect of METTL3 on EV activity? Does this also contribute to the effect of METTL3 on wild-type reporter? This need to be addressed.

We thank the reviewer for this viewpoint. We acknowledged that the original figure we showed may cause ambiguity in understanding the data sets. To address this concern, we repeated this experiment and pooled all data from three new independent experiments. Our new data showed that compared with EV-Myc: EV group, METTL3-Myc only slightly but not significantly increased the luciferase activity of EV (**Figure 6k**). We assumed that potential m⁶A sites existed in EV might attribute to the luciferase activity. However, these data indicated that the m⁶A site existed in the *Tcf7* indeed augmented the luciferase activity, supporting the conclusion that abrogation of m⁶A activity is associated with decreased expression level of TCF-1.

4. Figure 6k and 6l: the effect of METTL3 deletion on the mRNA stability of *Tcf7* is moderate. Can other mechanism be involved in the regulation of *Tcf7* stability and expression by METTL3?

We thank the reviewer for his/her concern. Our data reflect the mRNA stability of *Tcf7* is influenced with METTL3 deletion. However, the difference looks moderate is probably caused by the experimental system, depending on the features of m⁶A modification, cell type, and experimental condition *in vitro* (Huang et al., 2018; Li et al., 2017; Paris et al., 2019; Weng et al., 2018; Wu et al., 2019). Meanwhile, we agreed with the reviewer, we cannot exclude other potential regulatory layers and draw a conclusion that the METTL3 deletion on the mRNA stability of *Tcf7* is the unique mechanism due to the limitation of current results and our knowledge. It is will be of great interest to study other potential mechanisms of m⁶A modification on *Tcf7* stability in the future study.

5. Does METTL3 interact with the *Tcf7* transcripts?

We thank the reviewer for giving this constructive suggestion. As requested, we performed RNA-IP assay by using an antibody against METTL3 in CD4⁺ T cells. The results indicated that METTL3 could directly interact with the *Tcf7* transcripts (**Figure 6g**).

6. The effect of METTL3 deletion on m⁶A enrichment in the *Tcf7* gene needs to be validated by m⁶A IP qPCR analysis.

We agreed with the reviewer for this concern. Actually, the m⁶A-RIP-qPCR result has been shown in original Figure 6g, illustrating METTL3 deletion impairs the m⁶A enrichment on *Tcf7* gene. In revised manuscript, we reorganized the entire data set and presented the relative data in current **Figure 6h**. For your convenience, the result is also shown as follow (Figure R7).

Figure R7. Analysis of m⁶A enrichment on *Tcf7* mRNA by m⁶A-RIP-qPCR

7. The importance of TFH cell differentiation regulated by METTL3 can be further assessed. For example, does adoptive transfer of TFH cells from WT mice rescue the defect in CD4-specific METTL3 deletion mice in response to viral infection?

We thank the reviewer for giving us this constructive suggestion, which was also raised by the Reviewer 1 (major concern 4). To address this point, we adoptively transferred wild-type SMARTA cells into Ctrl and *Mettl3^{fl/fl}Cd4-Cre* host mice. We found the cell numbers of both GC B cells and plasma cells in *Mettl3^{fl/fl}Cd4-Cre* mice were comparable with Ctrl mice, in parallel with the GC areas in the spleens. Meanwhile, the level of antigen-specific IgG in *Mettl3^{fl/fl}Cd4-Cre* host mice (transferred with SMARTA cells) showed no obvious difference with Ctrl host mice, while *Mettl3^{fl/fl}Cd4-Cre* mice (no SMARTA cells transfer) still exhibited profoundly defects in the production of LCMV-specific IgG (**Figure 2d-g**). These results strongly indicated transfer of T_{FH} cells from WT mice could rescue the defects in CD4-specific METTL3 deletion mice in response to viral infection.

Minor comments:

1. Line 264: typo in “MRTTL3”

We thank the reviewer for this correction and we have modified this spelling mistake in revised version, accordingly.

2. Line 302-303: sentence needs correction. “Does is....”.

We thank the reviewer for pointing out this mistake and we have modified it as “Does METTL3-mediated T_{FH} differentiation also directly depend on its m⁶A catalytic activity?”.

Reference

- Barbet, G., Sander, L.E., Geswell, M., Leonardi, I., Cerutti, A., Iliiev, I., and Blander, J.M. (2018). Sensing Microbial Viability through Bacterial RNA Augments T Follicular Helper Cell and Antibody Responses. *Immunity* 48, 584-598 e585.
- Choi, Y.S., Gullicksrud, J.A., Xing, S., Zeng, Z., Shan, Q., Li, F., Love, P.E., Peng, W., Xue, H.H., and Crotty, S. (2015). LEF-1 and TCF-1 orchestrate T_{FH} differentiation by regulating differentiation circuits upstream of the transcriptional repressor Bcl6. *Nat. Immunol.* 16, 980-990.
- Choi, Y.S., Kageyama, R., Eto, D., Escobar, T.C., Johnston, R.J., Monticelli, L., Lao, C., and Crotty, S. (2011). ICOS receptor instructs T follicular helper cell versus effector cell differentiation via induction of the transcriptional repressor Bcl6. *Immunity* 34, 932-946.
- Hale, J.S., Youngblood, B., Latner, D.R., Mohammed, A.U., Ye, L., Akondy, R.S., Wu, T., Iyer, S.S., and Ahmed, R. (2013). Distinct memory CD4⁺ T cells with commitment to T follicular helper- and T helper 1-cell lineages are generated after acute viral infection. *Immunity* 38, 805-817.
- Hao, Y., Wang, Y., Liu, X., Yang, X., Wang, P., Tian, Q., Bai, Q., Chen, X., Li, Z., Wu, J., et al. (2018). The Kinase Complex mTOR Complex 2 Promotes the Follicular Migration and Functional Maturation of Differentiated Follicular Helper CD4⁺ T Cells During Viral Infection. *Front. Immunol.* 9, 1127.
- Huang, H., Weng, H., Sun, W., Qin, X., Shi, H., Wu, H., Zhao, B.S., Mesquita, A., Liu, C., Yuan, C.L., et al. (2018). Recognition of RNA N⁶-methyladenosine by IGF2BP proteins enhances mRNA stability and translation. *Nat. Cell Biol.* 20, 285-295.
- Li, F., Zeng, Z., Xing, S., Gullicksrud, J.A., Shan, Q., Choi, J., Badovinac, V.P., Crotty, S., Peng, W., and Xue, H.H. (2018). Ezh2 programs T_{FH} differentiation by integrating phosphorylation-dependent activation of Bcl6 and polycomb-dependent repression of p19Arf. *Nat. Commun.* 9, 5452.
- Li, H.B., Tong, J., Zhu, S., Batista, P.J., Duffy, E.E., Zhao, J., Bailis, W., Cao, G., Kroehling, L., Chen, Y., et al. (2017). m⁶A mRNA methylation controls T cell homeostasis by targeting the IL-7/STAT5/SOCS pathways. *Nature* 548, 338-342.
- Marshall, H.D., Chande, A., Jung, Y.W., Meng, H., Poholek, A.C., Parish, I.A., Rutishauser, R., Cui, W., Kleinstein, S.H., Craft, J., et al. (2011). Differential expression of Ly6C and T-bet distinguish effector and memory Th1 CD4⁺ cell properties during viral infection. *Immunity* 35, 633-646.
- Nish, S.A., Zens, K.D., Kratchmarov, R., Lin, W.W., Adams, W.C., Chen, Y.H., Yen, B., Rothman, N.J., Bhandoola, A., Xue, H.H., et al. (2017). CD4⁺ T cell effector commitment coupled to self-renewal by asymmetric cell divisions. *J. Exp. Med.* 214, 39-47.
- Paris, J., Morgan, M., Campos, J., Spencer, G.J., Shmakova, A., Ivanova, I., Mapperley, C., Lawson, H., Wotherspoon, D.A., Sepulveda, C., et al. (2019). Targeting the RNA m⁶A reader YTHDF2 selectively compromises cancer stem cells in acute myeloid leukemia. *Cell Stem Cell* 25, 137-148 e136.
- Ramirez, F., Ryan, D.P., Gruning, B., Bhardwaj, V., Kilpert, F., Richter, A.S., Heyne, S., Dundar, F., and Manke, T. (2016). deepTools2: a next generation web server for deep-sequencing data analysis. *Nucleic Acids Res.* 44, W160-165.
- Tong, J., Cao, G., Zhang, T., Sefik, E., Amezcua Vesely, M.C., Broughton, J.P., Zhu, S., Li, H., Li, B., Chen, L., et al. (2018). m⁶A mRNA methylation sustains Treg suppressive functions. *Cell Res.* 28, 253-256.
- Vaeth, M., Eckstein, M., Shaw, P.J., Kozhaya, L., Yang, J., Berberich-Siebelt, F., Clancy, R., Unutmaz, D., and Feske, S. (2016). Store-Operated Ca²⁺ Entry in Follicular T Cells Controls Humoral Immune Responses and Autoimmunity. *Immunity* 44, 1350-1364.
- van Loosdregt, J., Fleskens, V., Tiemessen, M.M., Mokry, M., van Boxtel, R., Meerding, J., Pals, C.E., Kurek, D., Baert, M.R., Delemarre, E.M., et al. (2013). Canonical Wnt signaling negatively modulates regulatory T cell function. *Immunity* 39, 298-310.
- Vinuesa, C.G., Cook, M.C., Angelucci, C., Athanasopoulos, V., Rui, L., Hill, K.M., Yu, D., Domaschenz, H., Whittle, B., Lambe, T., et al. (2005). A RING-type ubiquitin ligase family member required to repress follicular helper T cells and autoimmunity. *Nature* 435, 452-458.
- Weng, H., Huang, H., Wu, H., Qin, X., Zhao, B.S., Dong, L., Shi, H., Skibbe, J., Shen, C., Hu, C., et al. (2018). METTL14 inhibits hematopoietic stem/progenitor differentiation and promotes leukemogenesis via mRNA m⁶A modification. *Cell Stem Cell* 22, 191-205 e199.
- Wu, R., Li, A., Sun, B., Sun, J.G., Zhang, J., Zhang, T., Chen, Y., Xiao, Y., Gao, Y., Zhang, Q., et al. (2019). A novel m⁶A reader Prrc2a controls oligodendroglial specification and myelination. *Cell Res.* 29, 23-41.
- Wu, T., Shin, H.M., Moseman, E.A., Ji, Y., Huang, B., Harly, C., Sen, J.M., Berg, L.J., Gattinoni, L., McGavern, D.B., et al. (2015). TCF1 is required for the T follicular helper cell response to viral infection. *Cell Rep.* 12, 2099-2110.

- Xing, S., Gai, K., Li, X., Shao, P., Zeng, Z., Zhao, X., Zhao, X., Chen, X., Paradee, W.J., Meyerholz, D.K., et al. (2019). Tcf1 and Lef1 are required for the immunosuppressive function of regulatory T cells. *J. Exp. Med.* *216*, 847-866.
- Xu, L., Cao, Y., Xie, Z., Huang, Q., Bai, Q., Yang, X., He, R., Hao, Y., Wang, H., Zhao, T., et al. (2015). The transcription factor TCF-1 initiates the differentiation of T_{FH} cells during acute viral infection. *Nat. Immunol.* *16*, 991-999.
- Yang, B.H., Wang, K., Wan, S., Liang, Y., Yuan, X., Dong, Y., Cho, S., Xu, W., Jepsen, K., Feng, G.S., et al. (2019). TCF1 and LEF1 Control Treg Competitive Survival and Tfr Development to Prevent Autoimmune Diseases. *Cell Rep.* *27*, 3629-3645 e3626.
- Yu, Q., Sharma, A., Oh, S.Y., Moon, H.G., Hossain, M.Z., Salay, T.M., Leeds, K.E., Du, H., Wu, B., Waterman, M.L., et al. (2009). T cell factor 1 initiates the T helper type 2 fate by inducing the transcription factor GATA-3 and repressing interferon-gamma. *Nat. Immunol.* *10*, 992-999.
- Zhang, J., He, Z., Sen, S., Wang, F., Zhang, Q., and Sun, Z. (2018). TCF-1 Inhibits IL-17 Gene Expression To Restrain Th17 Immunity in a Stage-Specific Manner. *J. Immunol.* *200*, 3397-3406.
- Zhu, Y., Zhao, Y., Zou, L., Zhang, D., Aki, D., and Liu, Y.C. (2019). The E3 ligase VHL promotes follicular helper T cell differentiation via glycolytic-epigenetic control. *J. Exp. Med.* *216*, 1664-1681.

REVIEWERS' COMMENTS

Reviewer #1 (Remarks to the Author):

All question were answered to full satisfaction.
Thank you!

Reviewer #2 (Remarks to the Author):

The authors have addressed my comments as well as those of the other reviewers. Just two small comments:

1) if they could state that the Tcf7 retrovirus does not have the 3' untranslated region, that would be helpful.

2) The authors state: "the expression of Tcf7 was intensely lower in METTL3-deficient TFH cells than in wild-type cells". Tcf7 is not more significantly lower than many of the other factors they examined in this figure. Perhaps they could just state: that among the genes with low expression, Tcf7 was of interest because of its role in Tfh cell differentiation. (or both Bcl6 and Tcf7, but then you argue later why you are interested in Tcf7.

Reviewer #3 (Remarks to the Author):

The authors worked very hard, added multiple new data to address the comments overall. Now the manuscript has improved significantly. I have no further comments.

Point-by-point response to reviewers

Reviewer #1 (Remarks to the Author):

All question were answered to full satisfaction.

Thank you!

We thank this reviewer again for his/her professional evaluation on work and for giving us constructive suggestions to improve the quality of our MS during the review process.

Reviewer #2 (Remarks to the Author):

The authors have addressed my comments as well as those of the other reviewers. Just two small comments:

We highly appreciate the reviewer for his/her meditation of our responses and modified our manuscript according to the suggestions.

1) if they could state that the *Tcf7* retrovirus does not have the 3' untranslated region, that would be helpful.

We thank the reviewer for this suggestion. To better characterize the strategy of retroviral transduction of *Tcf7*, we modified our statement as "Upon transducing in vivo primed SMARTA CD4⁺ T cells with TCF-1 (full length CDS of P45 isoform without 3' untranslated region) retrovirus" (Page 10, Lines 211-214).

2) The authors state: "the expression of *Tcf7* was intensely lower in METTL3-deficient TFH cells than in wild-type cells". *Tcf7* is not more significantly lower than many of the other factors they examined in this figure. Perhaps they could just state: that among the genes with low expression, *Tcf7* was of interest because of its role in Tfh cell differentiation. (or both *Bcl6* and *Tcf7*, but then you argue later why you are interested in *Tcf7*).

We thank the reviewer for this constructive suggestion. To illustrate our statement more precisely, we modified the sentence referred by the reviewer as "In addition, the expression of *Tcf7* was significantly lower in METTL3-deficient T_{FH} cells than in wild-type T_{FH} cells (Fig. 4e). Among the genes with low expression, *Tcf7* was of interest because of its role in Tfh cell differentiation, so we also performed GSEA analysis of a gene set containing TCF-1-activated genes in T_{FH} cells" (Page 13, Lines 296-297).

Reviewer #3 (Remarks to the Author):

The authors worked very hard, added multiple new data to address the comments overall. Now the manuscript has improved significantly. I have no further comments.

We thank the reviewer for the final acceptance of our manuscript.